# Two distinct catalytic pathways for GH43 xylanolytic enzymes unveiled by X-ray and QM/MM simulations

Mariana A. B. Morais [1,2], Joan Coines [2], Mariane N. Domingues [1], Renan A. S. Pirolla[1], Celisa C. C. Tonoli[3], Camila R. Santos[1], Jessica B. L. Correa[1], Fabio C. Gozzo [4], Carme Rovira [2,5✉] & Mario T. Murakami [1✉]

Xylanolytic enzymes from glycoside hydrolase family 43 (GH43) are involved in the break-down of hemicellulose, the second most abundant carbohydrate in plants. Here, we kinetically and mechanistically describe the non-reducing-end xylose-releasing exo-oligoxylanase activity and report the crystal structure of a native GH43 Michaelis complex with its substrate prior to hydrolysis. Two distinct calcium-stabilized conformations of the active site xylosyl unit are found, suggesting two alternative catalytic routes. These results are confirmed by QM/MM simulations that unveil the complete hydrolysis mechanism and identify two possible reaction pathways, involving different transition state conformations for the cleavage of xylooligosaccharides. Such catalytic conformational promiscuity in glycosidases is related to the open architecture of the active site and thus might be extended to other exo-acting enzymes. These findings expand the current general model of catalytic mechanism of glycosidases, a main reaction in nature, and impact on our understanding about their interaction with substrates and inhibitors.

[1] Brazilian Biorenewables National Laboratory (LNBR), Brazilian Center for Research in Energy and Materials (CNPEM), Campinas 13083-100, Brazil. [2] Departament de Química Inorgànica i Orgànica & Institut de Química Teòrica i Computacional (IQTCUB), Universitat de Barcelona, Barcelona 08028, Spain. [3] Brazilian Biosciences National Laboratory (LNBio), Brazilian Center for Research in Energy and Materials (CNPEM), Campinas 13083-100, Brazil. [4] Dalton Mass Spectrometry Laboratory, Institute of Chemistry, University of Campinas, Campinas 13083-970, Brazil. [5] Institució Catalana de Recerca i Estudis Avançats (ICREA), Barcelona 08010, Spain. ✉email: c.rovira@ub.edu; mario.murakami@lnbr.cnpem.br

Hemicellulose, a main component of plant cell walls, is one of the most abundant and complex carbohydrates in nature, composed of several sugars including mostly β-xylose and α-arabinose. Microorganisms mastering plant cell wall degradation are provided with multiple genes with broad action on hemicellulose, commonly encoding several glycoside hydrolases from family 43 (GH43)[1,2]. They constitute one of the largest GH families annotated in the CAZy database[3] and are divided into 37 subfamilies that diverge in terms of structure and function[4]. A significant fraction of these subfamilies remains partially characterized without structural data, complete functional profile or even fully unexplored. In addition, due to their atypical diverse polysaccharide specificity, members from the same subfamily might exhibit distinct modes of action, substrate recognition, and quaternary structure[5,6].

A striking feature of GH43 enzymes is their bi-functionality, with several members exhibiting both β-xylosidase and α-arabinofuranosidase activities[7–11]. Taking advantage of their intrinsic promiscuous scaffold, alternative functions have been discovered or introduced through rational redesign of the active-site cleft[6,12–14], which make members of this family good candidates for engineering relevant modes of action for industrial processes. To further increase the biotechnological potential of GH43 enzymes and to better understand their biological roles in microbial communities specialized in plant cell wall degradation, it is instrumental to elucidate their mechanisms of action and catalytic pathways at molecular detail, which have been so far not revealed.

GH43 enzymes cleave glycosidic bonds with inversion of the anomeric configuration. The reaction is catalyzed by two essential carboxylate-based residues (glutamic or aspartic acid). One of them (the general acid) protonates the scissile glycosidic oxygen atom, whereas the other essential residue (the general base) coordinates the nucleophile, a water molecule, to assist its deprotonation to complete the reaction (Fig. 1). Based on crystallographic data, some works suggested the importance of calcium for maintaining the protonation state of the catalytic general acid residue through the interaction with a water molecule[15,16]. In fact, calcium ions have been found to be essential for the catalytic activity of GH43 enzymes (arabinanases, xylosidases, and arabinofuranosidases)[14,15,17–19]. They are observed in several crystal structures in a hydrated cavity located near the −1 subsite, but not directly interacting with the substrate[12,14–17,20–22]. However, the molecular basis behind the role of calcium in GH43 enzymes remains an intriguing and unanswered question.

Another open question in GH43 catalytic mechanisms concerns the conformation that the substrate follows during the hydrolysis reaction, in particular the conformational itinerary of

the sugar located at the −1 enzyme subsite (the one bearing the scissile glycosidic bond)[23]. In glycosidases, such itinerary is a signature of enzyme family, and has been proven to be instrumental for inhibitor and activity-based probe design[24,25]. In the case of GH43 enzymes, those acting on β-galactans have been proposed to follow a $^1S_3 \rightarrow [^4H_3]^{\ddagger} \rightarrow {}^4C_1$ itinerary[24,26], whereas a $^2S_O \rightarrow [^{2,5}B]^{\ddagger} \rightarrow {}^5S_1$ itinerary was predicted for those acting on β-xylose derivatives[27]. Considering the large functional diversity within GH43 family, with specificity for distinct polysaccharides or sugar moieties, the conformational routes and catalytic mechanisms of GH43 enzymes might further diverge. However, the rarity of Michaelis complex structures available for these enzymes hampers the elucidation of their molecular mechanisms of action.

Here we report the non-reducing-end xylose-releasing exo-oligoxylanase activity in the GH43 family and provide the crystal structure of a native GH43 Michaelis complex prior to catalysis. The two saccharide configurations trapped in the active Michaelis complex showed to be catalytically viable through distinct conformational itineraries as demonstrated by QM/MM metadynamics. The placing of catalytic conformational promiscuity in glycosidases, a major class of enzymes in most living systems and of high biotechnological relevance, will impact on the current understanding of how these enzymes can interact with substrates and inhibitors. Moreover, we demonstrate the calcium role in the GH43 activity, which involves the stabilization of the productive configuration of the enzyme-substrate complex including the pre-activated −1 saccharide conformation and the catalytically competent state of the general base.

## Results

**Discovery of a GH43 calcium-activated exo-oligoxylanase.** The xylanolytic system of *Xanthomonas citri* pv. *citri* (Xac) is highly equipped with several CAZymes involved in xylan breakdown, including three GH10s, one GH51, one GH39, two predicted GH30, and six GH43s. Some of these enzymes exhibited atypical functional attributes such as the cleavage of the recalcitrant internal di-substitutions in arabinoxylans and the reducing-end exo-oligo activity[28,29]. Herein, we investigated another enzyme from this system, encoded by the ORF Xac4258 (named here as XacGH43_1), which displayed both β-xylosidase and α-arabinofuranosidase activities, with preference for xylose-derived substrates, according to activity assays against several natural and synthetic substrates (Supplementary Table 1). The enzyme shows typical pH and temperature dependence observed for other CAZymes from this phytopathogen (Supplementary Fig. 1) and capillary zone electrophoresis (CZE) experiments revealed that XacGH43_1 releases xylose from all xylooligo-saccharides (XOS) tested (from xylobiose (X2) to xylohexaose (X6)) (Fig. 2a). Its catalytic activity is positively influenced by the presence of $Ca^{2+}$ ions, in comparison to the control and to other cations (Fig. 2b), being also responsive to increasing concentrations of $CaCl_2$ (Fig. 2b *inset*). To verify the preference of XacGH43_1 for longer or shorter XOS, kinetics assays were performed using quantitative mass spectrometry to monitor the xylose release from X2 and X6 (Fig. 2c, d). Kinetic parameters revealed ≈threefold higher catalytic efficiency of XacGH43_1 on X6 compared to X2, without the addition of $CaCl_2$, and ≈eightfold higher when $CaCl_2$ was added to the reaction (Fig. 2c, d and Supplementary Table 2). The saturation over xylan was not reached, indicating that XacGH43_1 is able to cleave efficiently only XOS, despite being active on the polymeric substrate (Supplementary Fig. 1).

The calcium was found to be responsible for improving the substrate affinity ($K_M$) in one order of magnitude when tested

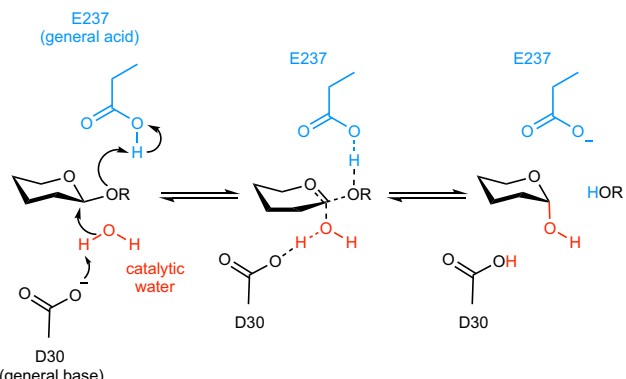

**Fig. 1 General reaction mechanism of inverting GH43 enzymes.** Residues numbers correspond to the enzyme under investigation.

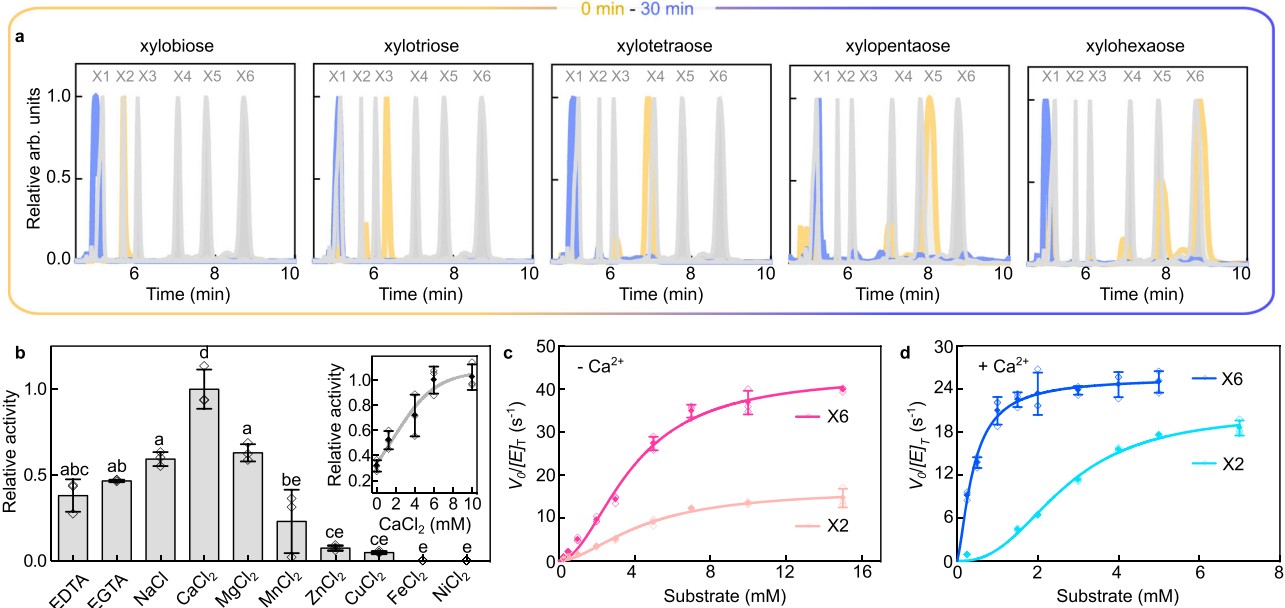

**Fig. 2 Catalytic properties of XacGH43_1. a** Capillary electrophoresis analysis showing the cleavage pattern of xylobiose, xylotriose, xylotetraose, xylopentaose, and xylohexaose. The reactions were interrupted at 0 min (dark yellow) or 30 min (blue). The standard patterns of xylose (X1), xylobiose (X2), xylotriose (X3), xylotetraose (X4), xylopentaose (X5), and xylohexaose (X6) are shown as gray curves. All data were normalized (starting from 4.5 min to omit APTS signal). **b** Influence of cations and chelating agents in XacGH43_1 activity and the enzyme response with increasing concentrations of $CaCl_2$ (inset). Results are expressed as mean ± SD from three independent experiments. Data points are shown as empty symbols. In the bars plot, one-way ANOVA ($P < 0.05$) and tukey post hoc tests were performed for multiple comparisons ($P < 0.001$). Bars showing the same letters do not present significant statistical difference, according to tukey post hoc test. **c** Kinetics curves of XacGH43_1 activity on X2 (light pink) or X6 (dark pink) without the addition of $CaCl_2$ in the reaction. **d** Kinetics curves of XacGH43_1 activity on X2 (light blue) or X6 (dark blue) with the addition of 6 mM $CaCl_2$ in the reaction. The (**c, d**) curves were built based on mass spectrometric analysis of xylose release by the enzyme and data are expressed as mean ± SD from three independent experiments ($V_0$ = initial velocity; $[E_T]$ = enzyme concentration). Data points are shown as empty symbols. Source data are provided as a source data file.

with the preferred substrate (X6). The catalytic turnover ($k_{cat}$), even though, was not affected as the $K_M$, showing that the calcium influence on the catalytic efficiency was due to the increase in substrate affinity. All these observations indicate that we have found a calcium-dependent xylose-releasing exo-oligoxylanase.

**A crystallographic snapshot of an active Michaelis substrate complex**. To understand the catalytic route for the exo-oligoxylanase activity in the GH43 family, the native enzyme and the complex with product (xylose) were crystallized. In addition, a complex with the substrate (xylooligosaccharide) was obtained by means of short soaking followed by flash-freezing in a nitrogen stream[30].

The XacGH43_1 structure consists of a 5-bladed β-propeller catalytic core conserved in GH43 enzymes[31] and does not display accessory domain, which is observed in some GH43 subfamilies. According to structural comparisons, Asp30 acts as the general base, Glu237 as the general acid, and Asp150 as the pKa modulator, composing the GH43 invariant triad described by Brüx et al.[5] (Fig. 3a).

The structure of the free enzyme shows a glycerol molecule occupying the −1 subsite (Fig. 3a). In the complex with xylose, the monosaccharide was found occupying the −1 subsite, in a relaxed chair conformation ($^4C_1$), in the two molecules of the asymmetric unit (Fig. 3b). The complex of XacGH43_1 with xylotriose shows that the substrate is bound into the active site of one molecule of the asymmetric unit, occupying the subsites −1, +1, and +2 (Fig. 3c). As the xylotriose is uncleaved, this complex represents a rare active Michaelis substrate complex prior to hydrolysis. A xylose molecule in the −1 subsite is present in the

other molecule of the asymmetric unit, with similar conformation as in the xylose complex.

All structures show that the active site is clearly blocked at the negative subsites, having only the highly conserved −1 subsite, which supports the exo mode of action, with the recognition of the non-reducing ends of the substrate. Molecular docking performed with X6 revealed that xylosyl residues beyond those crystallographic observed, i.e., +3, +4, and +5, can establish additional interactions with hydrophobic (Trp285, Trp344, Leu5, and Leu10) and polar (Thr312, Ser59, Gly310, and Gly309) residues from the very N- and C-termini (Supplementary Fig. 2). These residues act as an extended and continuous platform for substrate anchoring, which may explain the higher affinity to X6 compared to X2. Therefore, based on these functional and structural observations, XacGH43_1 can be defined as a GH43 non-reducing-end xylose-releasing exo-oligoxylanase.

**Calcium stabilizes a pre-catalytic configuration of the active site**. As described above, the crystal structure shows the presence of a $Ca^{2+}$ ion near the −1 subsite. The $Ca^{2+}$ ion is coordinated by the His288 $N_\epsilon2$ atom and by six water molecules in a pentagonal bipyramidal coordination geometry (Fig. 3 and Supplementary Fig. 3). The water molecules coordinating the calcium ion are connected to the carbonyl groups of residues Ser32, Asp150, Pro151, Ala99, Pro100, Asp30, His289, Gly238, and Pro239 (Supplementary Fig. 3). The catalytic triad, inferred from structural comparisons with other members of subfamily GH43_1[15,17], is not interacting with the calcium ion directly. In particular, residues Asp150 (pKa modulator) and Asp30 (general base) are part of the second coordination sphere of the calcium ion and

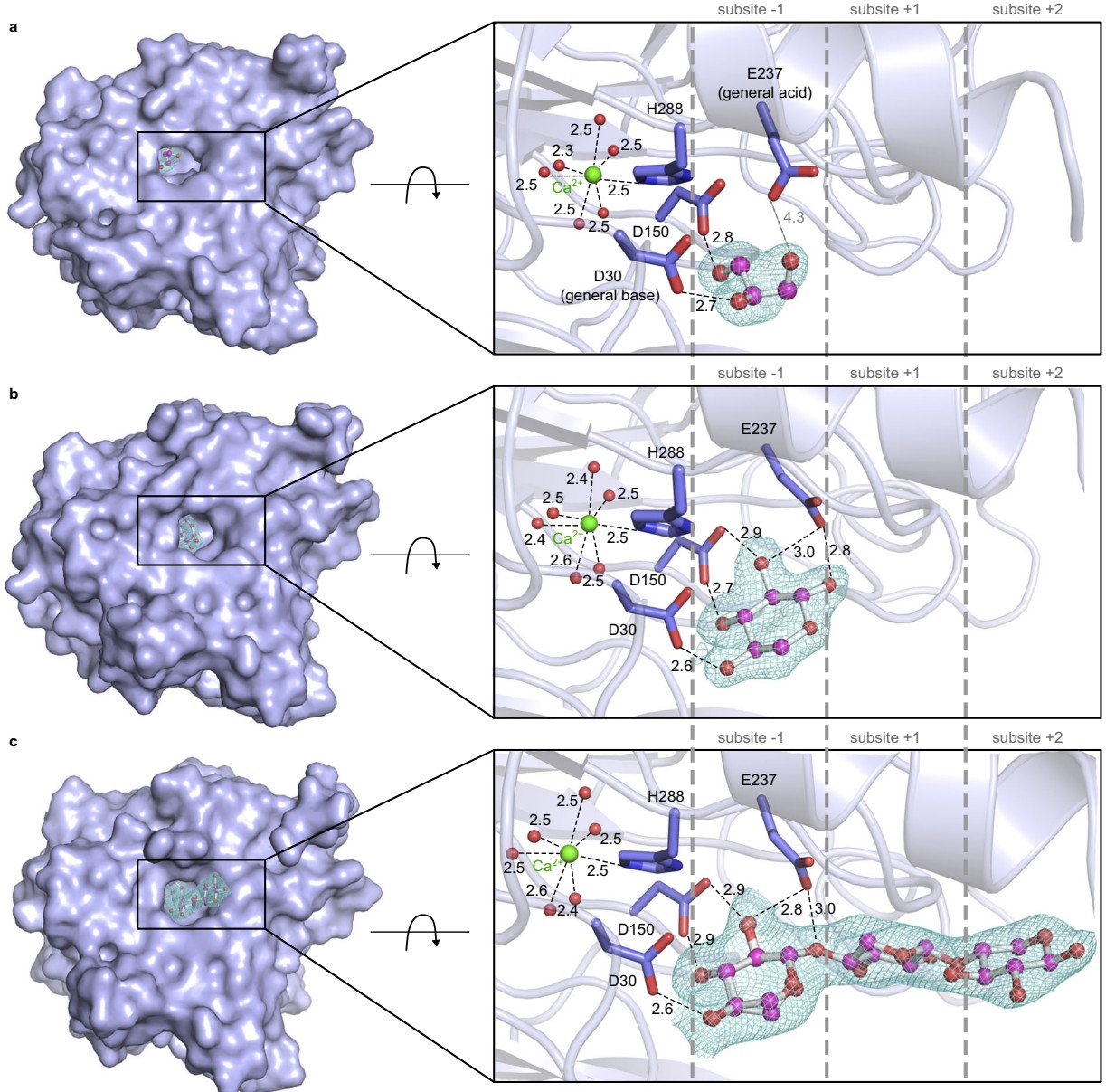

**Fig. 3 Crystal structures of XacGH43_1.** Structure of XacGH43_1 without sugar (**a**), where a glycerol molecule occupies the -1 subsite; in complex with xylose (**b**) and in complex with xylotriose (**c**). In all structures, the main interactions in the −1 subsite and with $Ca^{2+}$ (green sphere) are shown with distances in angstroms. Water molecules coordinating the $Ca^{2+}$ are also shown (red spheres). The $2F_o\text{-}F_c$ electron density maps of ligands are shown with contour level of 1.0 σ, after refinement. Subsites numbering scheme follows the previously proposed[91,92].

might interact with His288, while Glu237 (the general acid) is far from the $Ca^{2+}$ ion.

To further investigate how the $Ca^{2+}$ ion can affect enzyme stability and conformation, circular dichroism (CD) spectroscopy and small angle X-ray scattering (SAXS) were carried out. According to CD analysis, under calcium saturation, the enzyme had an increase of ≈5 °C in the transition temperature, indicating a gain of stability in the presence of the cation (Fig. 4a and Supplementary Fig. 4). SAXS data showed that the $Ca^{2+}$ ion was responsible for decreasing the protein flexibility (Fig. 4b) and the ab initio molecular envelope exhibited a low normalized spatial discrepancy with the crystallographic monomer in the presence of the cation (Fig. 4c, Supplementary Fig. 5), confirming the monomeric state of XacGH43_1 (+$Ca^{2+}$) in solution.

To assess the role of $Ca^{2+}$ ion in the structure and dynamics of the enzyme active site, we performed molecular dynamics (MD)

simulations considering both the WT enzyme ($Ca^{2+}$-bound) (Fig. 5a) and the enzyme in the absence of the $Ca^{2+}$ ion (in this case, we replaced it by a water molecule). Unexpectedly, in the absence of $Ca^{2+}$, a $Na^+$ ion from the solvent spontaneously entered into the calcium-binding site in just a few nanoseconds (Fig. 5b and Supplementary Fig. 6). While $Ca^{2+}$ ion was coordinated by six or seven water molecules during the entire simulation, $Na^+$ ion was coordinated by only four or five water molecules (Fig. 5a–c). This is in agreement with the sodium coordination previously observed by Matsuzawa et al.[15] for another calcium-activated GH43_1 enzyme in the absence of calcium[15]. In our simulations, the lack of an ion caused the side chain of His288 to alternate between two different orientations (named as rotamers 1 and 2, Fig. 5b, c; e, f), unlike what was observed in the $Ca^{2+}$-bound enzyme (rotamer 1). Moreover, the saccharide at the −1 subsite changed to an inverted chair

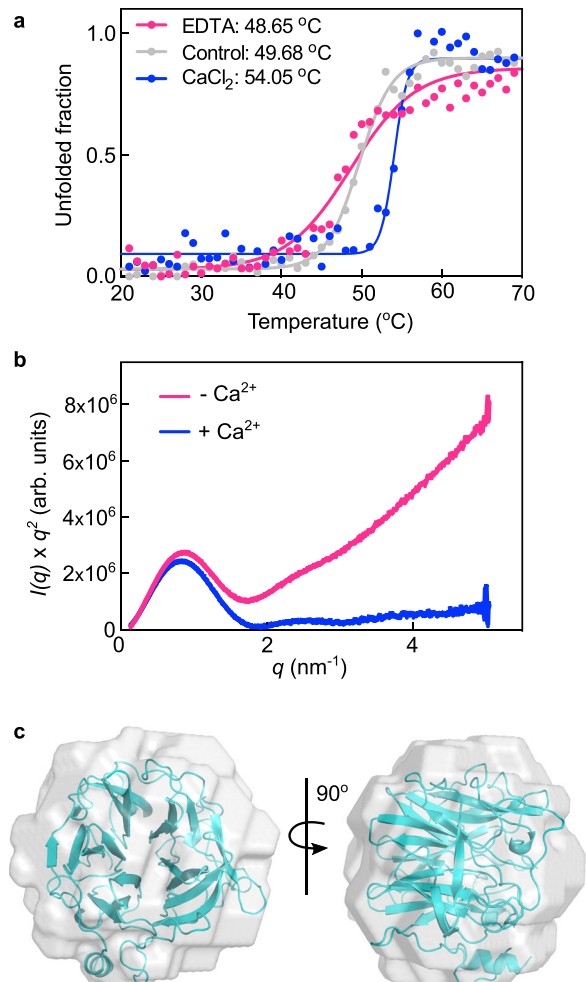

**Fig. 4 Ca$^{2+}$ increases XacGH43_1 stability. a** Thermal denaturation profile of XacGH43_1, monitored at 228 nm by circular dichroism spectroscopy without additives (control—gray), with EDTA (pink) or CaCl$_2$ (blue). The melting temperatures (T$_m$) are indicated for each sample. **b** Kratky plot[93] obtained for XacGH43_1 samples without (control—pink) or with CaCl$_2$ (blue). The dataset collected with CaCl$_2$ were used to generate an ab initio envelope. **c** XacGH43_1 crystal structure fitted into the SAXS envelope. Source data are provided as a source data file.

conformation ($^1C_4$), a high energy conformer for β-xylose[32] (see Fig. 5b, c and Supplementary Fig. 7a), a conformation that was not visited during the simulation of the Ca$^{2+}$-bound enzyme (Supplementary Fig. 7b). The lack of a divalent cation likely weakened the interaction between the Asp150 O$_\delta$2 atom and the xylosyl H2 that was instead attracted by Asp30, the general base, stabilizing the $^1C_4$ conformation (Fig. 5d) and breaking the interaction of Asp30 residue with the catalytic water. Furthermore, in the alternate rotameric conformation, the His288 N$_\delta$1 atom interacts with the general base (Asp30). All these changes disrupt active site catalytic configuration, weakening substrate binding when Ca$^{2+}$ is not present.

Previous experiments in the presence of Na$^+$ ions (even when not supplied in the reaction, Na$^+$ ions were present in the enzyme buffer; Fig. 3b) indicated a decrease in enzyme activity in comparison with Ca$^{2+}$ supplied samples, in consistence with the above findings. All these observations indicate a critical role of calcium-coordinated His288 in stabilizing a pre-activated conformation of the −1 xylosyl moiety and in maintaining a productive configuration of the enzyme catalytic machinery.

**Two putative catalytic pathways for the same substrate**. Interestingly, the Michaelis complex X-ray structure shows that the −1 xylopyranosyl ring of the xylotriose substrate adopts two distinct conformations, a $^4C_1$ chair and a distorted $^2S_O$ skew-boat, each one having 50% occupancy. This suggests that both conformations could contribute to the enzyme reaction mechanism. To get further insight into this question, we quantified the conformational flexibility of the substrate by computing the conformational free energy landscape (FEL) of the β-xylosyl unit at the −1 enzyme subsite with respect to the Cremer–Pople puckering coordinates[33], using QM/MM metadynamics. This approach has been used with success to predict the conformation of glycosides in the active site of several GHs[34–36].

The computed conformational FEL of the β-xylosyl ring, reconstructed from the metadynamics simulation, shows that the protein scaffold significantly restricts the conformations available for the −1 sugar so that only two main regions of low energy are accessible, a $^4C_1$ chair and a conformation in between $^2S_O$ and $^{2,5}B$ (Fig. 6a). Remarkably, the computed FEL is consistent with the two conformations observed in the crystal structure (Fig. 6b), both placed close to the minima of the conformational FEL. The small energetic difference between the two minima ($\approx$1 kcal mol$^{-1}$, in favor of the distorted conformation) explains why two different conformations are experimentally observed.

To elucidate the catalytic mechanism by which XacGH43_1 exo-oligoxylanase hydrolyzes XOS, we performed QM/MM metadynamics simulations of the hydrolytic reaction, starting from the native xylotriose Michaelis complex. Three collective variables (CVs), involving all covalent bonds to be cleaved by the enzyme, were used to drive the system from reactants (xylotriose) to products (xylose + xylobiose). The first collective variable (CV$_1$) accounts for proton transfer between Asp30 and the water molecule; CV$_2$ accounts for the nucleophilic attack of the catalytic water molecule; and CV$_3$ accounts for the transfer of the Glu237 proton to the glycosidic oxygen atom (Supplementary Fig. 8).

Of the two possible substrate conformations (Fig. 6), the distorted $^2S_O/^{2,5}B$ conformation is likely to be the most pre-activated for catalysis in view of its axial C1-O1' bond orientation[32]. It is thus expected that the enzyme preferentially reacts via this conformation. Accordingly, the simulations were started from a snapshot of the QM/MM MD simulation in which the −1 sugar is in a $^2S_O/^{2,5}B$ conformation.

Representative states along the minimum reaction free energy pathway are shown in Fig. 7a. Consistent with the conformational FEL (Fig. 6), the −1 sugar at the reactants state (R) adopts a conformation intermediate between $^2S_O$ and $^{2,5}B$. At this state, the catalytic water for the inverting mechanism is positioned at 1.80 ± 0.06 Å from the general base (Asp30) and 3.64 ± 0.07 Å from the anomeric carbon (Fig. 7b, c and Supplementary Table 3). The proton donor (general acid, Glu237) forms a 1.59 ± 0.38 Å hydrogen bond with the xylosyl O1' atom (Fig. 7b, c and Supplementary Table 3).

The reaction FEL obtained from the QM/MM metadynamics simulations, shown in Fig. 7d and Supplementary Fig. 9, is indicative of a concerted one-step reaction with two clear minima of low energy: the reactants (R), corresponding to the Michaelis complex and the products (P), separated by a single transition state (TS). The reaction free energy barrier (14.1 kcal mol$^{-1}$) is in agreement with the value estimated from the measured reaction rate ($\approx$15.8 kcal mol$^{-1}$).

The reaction begins with the elongation of the glycosidic bond, simultaneously with the transfer of the carboxylic hydrogen atom of the general acid residue to the glycosidic oxygen. From the reactants (R) to the TS, the −1 xylopyranosyl ring distorts from $^2S_O/^{2,5}B$ to $^{2,5}B$ (Fig. 7a), a conformation compatible with the

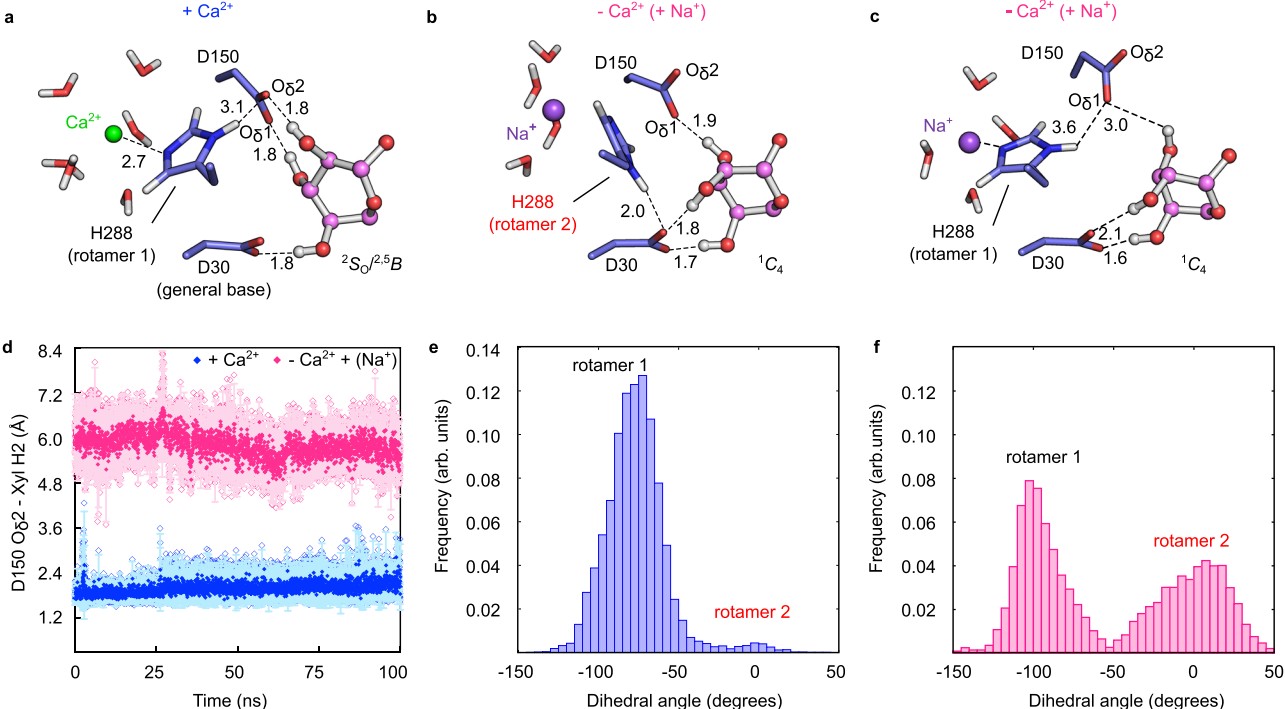

**Fig. 5 Effects of the presence or absence of Ca²⁺ ion in XacGH43_1 active site. a** Representation of the −1 subsite in the presence of $Ca^{2+}$ in the metal-binding site. **b** Snapshot of the MD simulation in which the $Ca^{2+}$ was initially replaced by a water molecule, followed by $Na^+$ (from the solvent) entrance to the metal-binding site. Here the His288 adopts a different conformation (rotamer 2) and interacts with Asp30 (general base) and Asp30 interacts with the xylosyl H2. **c** A snapshot from the same simulation of b, with the $Na^+$ being coordinated by His288. The interaction of Asp30-xylosyl H2 is maintained. Distances in **a**–**c** are represented in angstroms. **d** Distances between Asp150 $O_\delta 2$ and xylosyl H2 atoms during the production of the three MD independent simulations with $Ca^{2+}$ (blue) or with water (spontaneously replaced by $Na^+$ during the course of the simulation) (pink). Averages are shown as solid symbols and the SD with lighter colors. Data points are shown as empty symbols. **e** His288 dihedral angle distribution of the rotamers 1 and 2 during the MD simulation with $Ca^{2+}$. **f** Same results in the MD simulation without $Ca^{2+}$ (spontaneously replaced by $Na^+$ during the course of the simulation). Source data are provided as a source data file.

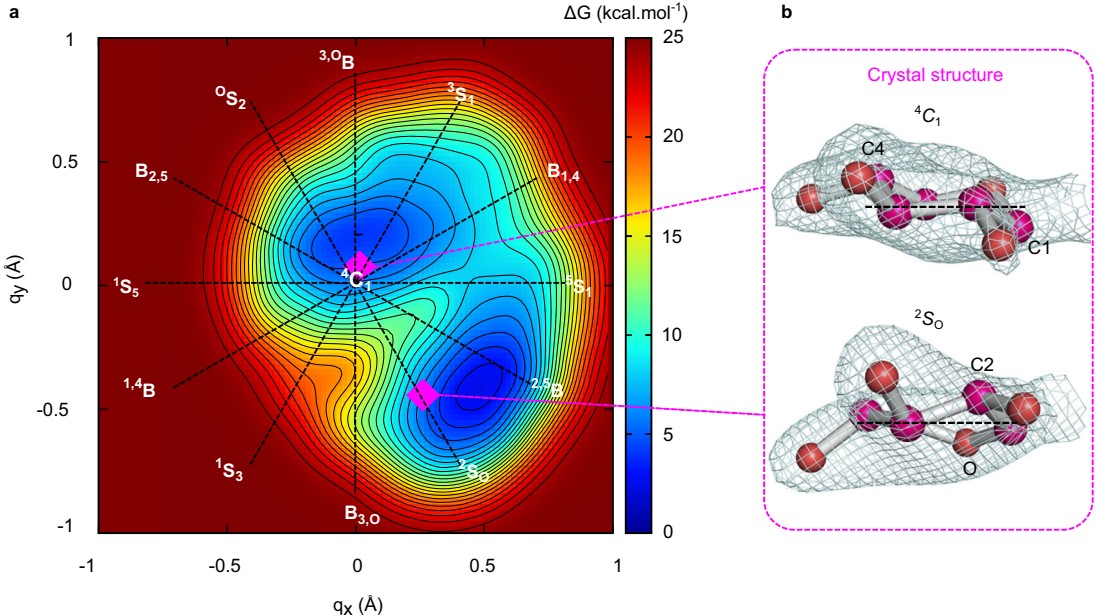

**Fig. 6 Conformations adopted by the xylopyranosyl ring at the −1 subsite of XacGH43_1 active site. a** FEL of xylopyranose in the −1 subsite of XacGH43_1 active-site obtained by ab initio metadynamics. The two conformations (found in the xylotriose crystal complex) are represented as purple squares. Isolines represent intervals of 1.0 kcal mol⁻¹. **b** Representation of the xylosyl residue at the −1 subsite in the xylotriose crystal complex, with 2F₀-Fᶜ electron density map contoured at 1.7 σ, after refinement. Source data are provided as a source data file.

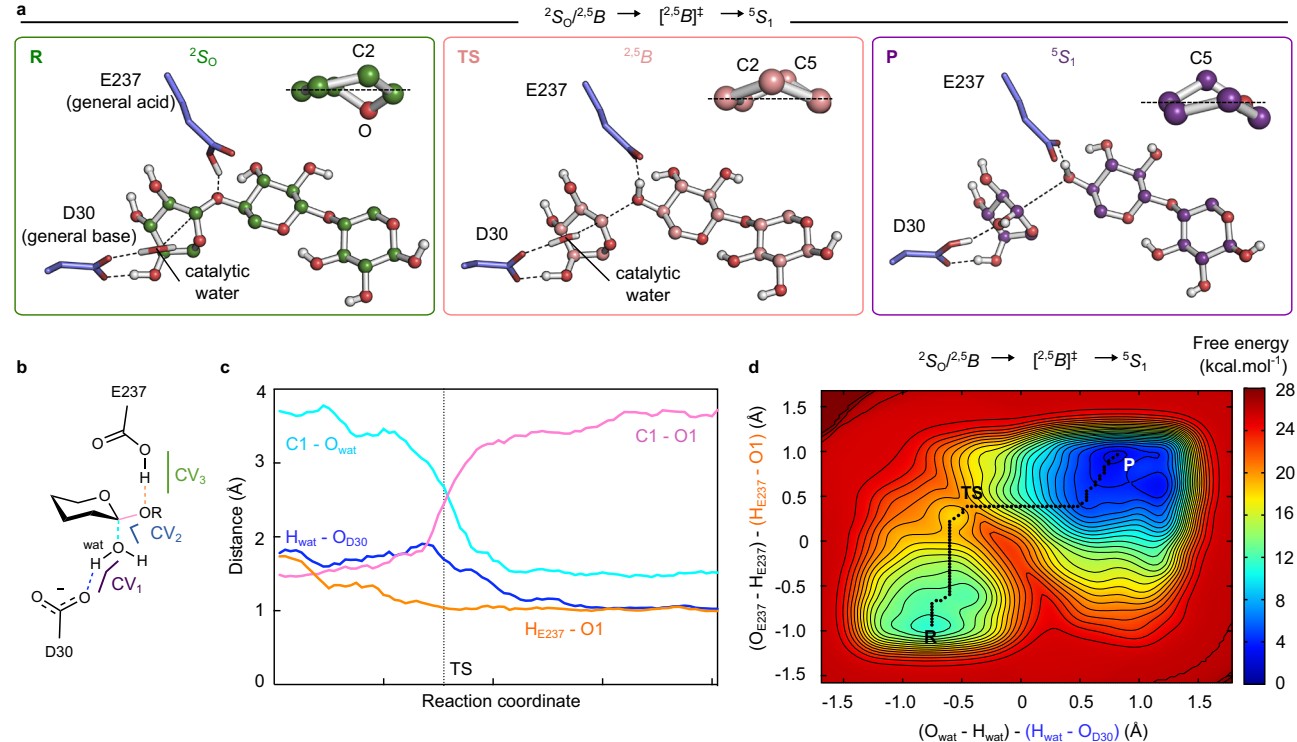

**Fig. 7 The $^{2}S_{O}/^{2,5}B \rightarrow [^{2,5}B]^{\ddagger} \rightarrow {}^{5}S_{1}$ catalytic itinerary. a** Representative snapshots of the reactant (R), transition state (TS) and product (P) states of the catalytic itinerary. A zoom of the −1 sugar conformation is represented at the top right of each state. **b** Schematic representation of catalytically relevant distances and collective variables (CV$_1$, CV$_2$, and CV$_3$) (wat = catalytic water). The CV$_1$ (purple) corresponds to the difference between O$_{wat}$–H$_{wat}$ and H$_{wat}$–O$_{Asp30}$ distances. The CV$_2$ (blue) was taken as the difference between C1$_{xylosyl}$–O4$_{xylosyl}$ and O$_{wat}$–C1$_{xylosyl}$ distances. The CV$_3$ (green) was taken as the difference between O$_{Glu237}$–H$_{Glu237}$ and H$_{Glu237}$–O1$_{xylosyl}$ distances. **c** Evolution of distances represented in **b** (running averages over five data points) along the reaction pathway. The distance between H$_{E237}$ – O1 is represented in orange, the distance between H$_{wat}$ – O$_{D30}$ in blue, C1-O$_{wat}$ in cyan and the glycosidic bond (C1-O1) in pink. **d** FEL of the catalytic reaction starting from $^{2}S_{O}/^{2,5}B$, projected into CV$_1$ ((O$_{wat}$–H$_{wat}$) − (H$_{wat}$–O$_{Asp30}$)) and CV$_3$ ((O$_{Glu237}$–H$_{Glu237}$) − (H$_{Glu237}$–O1$_{xylosyl}$)). The corresponding 3D FEL is shown in Supplementary Fig. 9. The minimum free energy pathway is indicated with a black dashed line on the two-dimensional FEL. Isolines are at intervals of 1 kcal mol$^{-1}$. Source data are provided as a source data file.

requirement of an oxocarbenium ion-like TS[23]. Likewise, the C1'-O5' bond of the −1 sugar shrinks with respect to its value at the reactants state (from $1.39 \pm 0.04$ to $1.29 \pm 0.02$ Å), indicative of the formation of a partial double bond between the C1' and O5' atoms. At the TS, the proton of the general acid residue (Glu237) is already transferred, the glycosidic bond is partially broken ($2.40 \pm 0.19$ Å) and the bond between the nucleophilic water and the anomeric carbon is partially formed ($2.72 \pm 0.15$ Å) (Fig. 7b, c and Supplementary Table 3). Proton transfer from the water to the general base residue (Asp30) takes place after the TS, while the −1 sugar changes to a $^{5}S_{1}$ conformation (P). Therefore, this conformational catalytic pathway of XacGH43_1 can be described as $^{2}S_{O}/^{2,5}B \rightarrow [^{2,5}B]^{\ddagger} \rightarrow {}^{5}S_{1}$. Interestingly, the conformation at P is in agreement with the one recently observed by Matsuzawa et al[15]. in the product complex structure (PDB ID 5GLN) of a β-xylosidase/α-L-arabinofuranosidase enzyme isolated from compost metagenome (CoXyl43), another member of the GH43_1 subfamily.

Another QM/MM metadynamics simulation was performed, following the same approach, starting from the −1 xylopyranosyl ring in the $^{4}C_{1}$ conformation, i.e., the alternative minimum in the conformational FEL that was also observed in the crystallographic complex. This conformation, with an equatorially oriented leaving group, is expected to be either unreactive or much less reactive than the distorted $^{2}S_{O}/^{2,5}B$ conformation, in which the leaving group is axial and the xyloside ring is pre-activated for catalysis[32]. Nevertheless, the evolution of relevant reaction

distances along the resulting reaction pathway $^{4}C_{1} \rightarrow [E_{3}]^{\ddagger} \rightarrow {}^{4}C_{1}$ (Fig. 8a) was found to be very similar to the previous itinerary (Fig. 8b, c and Supplementary Table 3). Moreover, the computed reaction FEL (Fig. 8d and Supplementary Fig. 10) is an indicative of a viable reaction pathway with a calculated free energy barrier of 17.5 kcal mol$^{-1}$, which is less favored than the one starting from a $^{2}S_{O}/^{2,5}B$ conformation but still compatible with experimental data. The reaction follows an unconventional itinerary for xylanolytic enzymes (and glycosidases in general) in which both Michaelis and product complexes have the same conformation ($^{4}C_{1}$), although the TS conformation ($E_{3}$) is fully compatible with an oxocarbenium ion-like species[23]. Therefore, both experiment and simulation show that the substrate can adopt two different conformations in the active site of XacGH43_1, each one leading to a distinct catalytic itinerary, $^{2}S_{O}/^{2,5}B \rightarrow [^{2,5}B]^{\ddagger} \rightarrow {}^{5}S_{1}$ and $^{4}C_{1} \rightarrow [E_{3}]^{\ddagger} \rightarrow {}^{4}C_{1}$.

Interestingly, one of the two itineraries here described for XacGH43_1, $^{2}S_{O}/^{2,5}B \rightarrow [^{2,5}B]^{\ddagger} \rightarrow {}^{5}S_{1}$, was previously described for a GH43 enzyme from another subfamily (GH43_11 from *Geobacillus stearothermophilus* T-6), which shares very low identity with XacGH43_1, indicating a conservation of the itinerary among GH43 subfamilies acting on xylose-derived substrates[27]. Even though alternative substrate conformations were not investigated by Barker and coworkers[27], the similarity of the active site cavity makes very likely that the conformational catalytic promiscuity obtained here are extensive to other GH43_11 members. These findings unfold an alternative

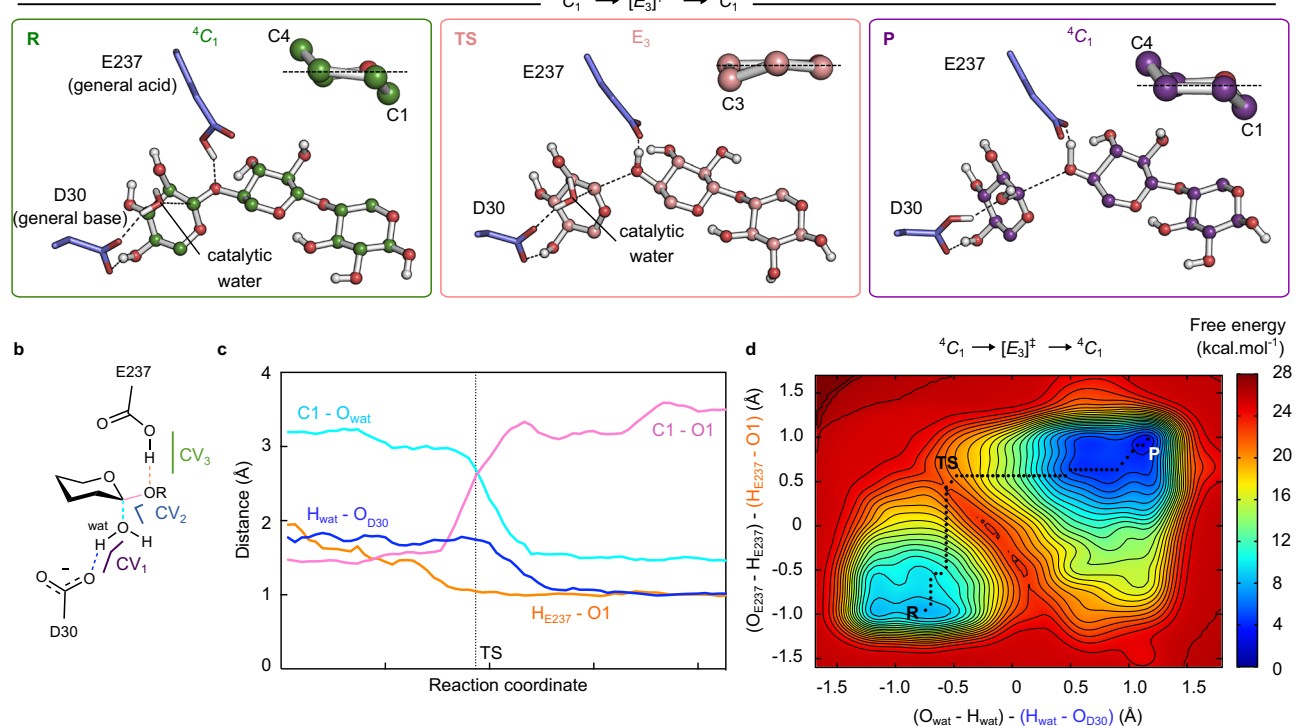

**Fig. 8 The $^4C_1 \rightarrow [E_3]^\ddagger \rightarrow {}^4C_1$ catalytic itinerary. a** Representative snapshots of the reactant (R) transition state (TS) and product (P) states of the catalytic itinerary. A zoom of the −1 sugar conformation is represented at the top right of each state. **b** Schematic representation of catalytically relevant distances and collective variables ($CV_1$, $CV_2$, and $CV_3$) (wat = catalytic water). The $CV_1$ (purple) corresponds to the difference between $O_{wat}$-$H_{wat}$ and $H_{wat}$-$O_{Asp30}$ distances. The $CV_2$ (blue) was taken as the difference between $C1_{xylosyl}$–$O4_{xylosyl}$ and $O_{wat}$–$C1_{xylosyl}$ distances. The $CV_3$ (green) was taken as the difference between $O_{Glu237}$-$H_{Glu237}$ and $H_{Glu237}$-$O1_{xylosyl}$ distances. **c** Evolution of these distances represented in **b** (running averages over five data points) along the reaction pathway. The distance between $H_{E237}$ – O1 is represented in orange, the distance between $H_{wat}$ – $O_{D30}$ in blue, C1-$O_{wat}$ in cyan and the glycosidic bond (C1-O1) in pink. **d** FEL of the catalytic reaction starting from $^4C_1$, projected into $CV_1$ (($O_{wat}$-$H_{wat}$) – ($H_{wat}$-$O_{Asp30}$)) and $CV_3$ (($O_{Glu237}$-$H_{Glu237}$) – ($H_{Glu237}$-$O1_{xylosyl}$)). The corresponding 3D FEL is shown in Supplementary Fig. 10. The minimum free energy pathway is indicated with a black dashed line on the two-dimensional FEL. Isolines are at intervals of 1 kcal mol⁻¹. Source data are provided as a source data file.

mechanistic aspect of glycosidases, in which two different catalytic itineraries might be viable for a given substrate.

## Discussion

The xylose-releasing exo-oligoxylanase activity has been kinetically and mechanistically demonstrated in the GH43 family. Such activity was characterized so far only in families GH8[37,38] and GH10[29] and in both cases the recognition mechanism involves the reducing ends, differing from that here reported for the GH43 family (non-reducing ends). Most GH43_1 enzymes were kinetically characterized in previous studies only against synthetic substrates or not systematically using different lengths of XOS from X2 to X6[39]. Thus, the exo-oligoxylanase activity could be possibly extended to other GH43_1 members.

As previously observed for other GH43 enzymes[14,15,18,19,22], $Ca^{2+}$ ions activate XacGH43_1. We verified that this cation is responsible for enhancing substrate affinity and for increasing protein stability, which has not been observed in other calcium-activated GH43 enzymes[14]. Our MD simulations show that the $Ca^{2+}$ ion is necessary for maintaining the active site in a favorable configuration for catalysis and that a $Na^+$ ion does not totally recover it. This is in accordance with the $Na^+$ ion impairing/decreasing activity in other GH43 enzymes in the absence of calcium[14,15]. Our MD simulations support that the $Ca^{2+}$ ion contributes to the binding of the substrate in three ways: (1) it stabilizes the proper conformation of the substrate at the −1 subsite; (2) it facilitates the interaction between the −1 sugar

and the pKa modulator Asp150; and (3) it controls the rotameric conformation of the catalytically relevant His288, preventing its interaction with the general base. This explains why mutations in the conserved active-site His impair the catalysis in calcium-activated GH43 members[22,40].

A snapshot of the Michaelis complex of the WT enzyme with a substrate (xylotriose) was obtained at high resolution. Such complexes are rarely available in GHs and the only precedent active Michaelis complex of a GH enzyme trapped in the crystal structure before hydrolysis was reported for a GH1 member[30]. The −1 xylosyl residue in the XacGH43_1 complex showed two alternative conformations, $^4C_1$ and $^2S_O$, which were further confirmed by QM/MM metadynamics simulations being almost iso-energetic.

QM/MM metadynamics simulations of the inverting reaction mechanism starting from the $^{2}S_O/^{2,5}B$ distorted conformation resulted in the catalytic itinerary $^{2}S_O/^{2,5}B \rightarrow [^{2,5}B]^\ddagger \rightarrow {}^5S_1$, which is in agreement with both the experimental free energy barrier and the trapped product of a cleaved X3 in the GH43_1 structure[15] (PDB ID 5GLN). In fact, this itinerary is similar to the one previously reported for another subfamily of GH43 enzymes ($^{2,5}B/^5S_1/B_{1,4} \rightarrow [^{2,5}B]^\ddagger \rightarrow {}^{2,5}B/^5S_1$)[27] that shares very low sequence identity with XacGH43_1, indicating that, despite differences in sequence and activity, this hydrolytic reaction mechanism is conserved in other GH43 subfamilies acting on XOS.

Although two alternative conformations of the −1 sugar were not expected to be catalytically competent, our results show that

the reaction starting from the $^4C_1$ conformation is also viable, with a free energy barrier just $\approx 3$ kcal mol$^{-1}$ higher than the one computed for the $^2S_O/^{2,5}B$ conformation. The reaction proceeds in this case via a different TS conformation ($E_3$), delineating a "cyclic" itinerary, $^4C_1 \rightarrow [E_3]^{\ddagger} \rightarrow {}^4C_1$. Therefore, although less favored, this reaction pathway cannot be excluded from contributing to the experimental reaction rate.

The fact that a chair conformation leads to a viable itinerary could be surprising a priori, since it has been shown that all β-GHs operate via a catalytic itinerary that begins from a specific distorted conformation of the sugar[24]. However, GH43_1 members are exo-acting enzymes, thus the leaving group can easily adapt to changes in the glycosidic bond during catalysis, in which an oxocarbenium ion-like is formed, something that it is more difficult in endo-acting GHs. In addition, these enzymes can bind distinct sugars in different stereochemistry, which may contribute to make them more likely to accept non-distorted XOS without imposing a notable decrease in enzyme performance. While this work was under review, another mechanistic study on a retaining exo-enzyme (the GH59 member β-galactocerebrosidase)[41] also pointed out to alternative and viable conformations of the −1 sugar and leaving group. This reinforces the concept of catalytic conformational promiscuity proposed here and indicates that it is not strictly linked to the substrate type and might occur in other exo-acting glycosidases independently of the stereochemical outcome (retaining or inverting).

In summary, beyond the molecular elucidation of calcium role and the high-resolution structure of an active Michaelis complex prior to catalysis, this work demonstrates that some glycosidases are more flexible than others in binding their substrate for catalysis. Conformational promiscuity may be relevant for the design of TS-like inhibitors and activity-based probes for GHs, opening other venues in glycobiology.

## Methods

**Cloning, protein production, and purification.** The sequence encoding XacGH43_1 (NCBI accession code AAM39093) was amplified from genomic DNA of Xac using the following nucleotides: 5′-CATATGTCCGATGAACTGCAAGCC-3′ and 5′-CTCGAG TCACCAAGGCGATACCGC−3′ and cloned into pET28a expression vector after digestion with NdeI and XhoI restriction enzymes. The recombinant XacGH43_1 was expressed in E. coli BL21 (DE3) cells. The culture was grown at 37 °C in Luria–Bertani medium 1% (w.v$^{-1}$) tryptone, 0.5% (w.v$^{-1}$) yeast extract, 1% (v.v$^{-1}$ sodium chloride) containing kanamycin (50 µg mL$^{-1}$) at 200 rpm until the O.D.$_{600\,nm}$ reached values of ~0.6 and incubated with 0.5 mM isopropyl-thio-D-galactopyranoside (Sigma-Aldrich, St. Louis, MO) for 16 h at 18 °C and 180 rpm. The cells were resuspended in lysis buffer (20 mM sodium phosphate, pH 7.4, 500 mM NaCl, 5 mM imidazole, and 4 mM PMSF) and incubated on ice with lysozyme (0.1 mg mL$^{-1}$) for 30 min and disrupted by sonication. The soluble extract was applied into a 5-mL HiTrap Chelating HP column (GE Healthcare, Little Chalfont, UK) previously charged with Ni$^{2+}$ and pre-equilibrated with 20 mM sodium phosphate, pH 7.4, 500 mM NaCl and 5 mM imidazole at a flow rate of 1 mL min$^{-1}$. XacGH43_1 was eluted using a non-linear gradient of imidazole (up to 0.5 M) at a flow rate of 1 mL min$^{-1}$. The eluted fractions were analyzed by SDS-PAGE[42], pooled, concentrated and submitted to size-exclusion chromatography in a HiLoad 16/600 Superdex 200 column (GE Healthcare), pre-equilibrated with 20 mM sodium phosphate, pH 7.4 and 150 mM NaCl at a flow rate of 1.0 mL min$^{-1}$. Samples from size-exclusion chromatography were analyzed by dynamic light scattering (DLS) in a Malvern ZetaSizer Nano series Nano-ZS (model ZEN3600) instrument (Malvern Zetasizer, Worcestershire, UK). DLS data were collected and analyzed with Zetasizer (7.12) software to evaluate sample homogeneity before pooling and concentration for crystallization trials.

**Circular dichroism spectroscopy.** Far UV CD spectra were recorded on a Jasco J-815 spectropolarimeter (Jasco International Co.,Tokyo, Japan) using a 1-mm quartz cuvette and Spectra Manager II software (Jasco). Samples at 7.5 µM in 20 mM Hepes pH 7.0 were used without additives (control) and with addition of 6 mM CaCl$_2$ or 5 mM EDTA. Spectra were collected at 20 °C, with a response time of 4 s nm$^{-1}$. Buffers spectra were also collected and subtracted from the respective sample. Thermal unfolding experiments were monitored at 228 nm. Samples were heated from 20 to 100 °C, with a heating rate of 1 °C min$^{-1}$. Melting temperatures were obtained by sigmoidal-Boltzman fit of denaturation curves from 20 to 85 °C.

**Small angle X-ray scattering (SAXS).** SAXS experiments were collected at protein concentration of 12 mg mL$^{-1}$ (300 µM) in 20 mM Hepes pH 7.5 with or without the addition of 6 mM CaCl$_2$. Measurements were performed using a 1-mm mica cell at the SAXS1 beamline from the Brazilian Synchrotron Light Laboratory (LNLS, Campinas, Brazil). Scattering data were recorded using a PILATUS 300 K (Dectris, Baden-Dattwil, Switzerland) and integrated using Fit2D[43]. Data were processed using the ATSAS package[44]. The program GNOM[45] was used to evaluate the pair-distance distribution function p(r). The ab initio molecular envelope was calculated for the sample with calcium using DAMMIN[46]. Averaged models were generated from several runs using DAMAVER[47]. The theoretical scattering curve of XacGH43_1 crystallographic coordinates was calculated using CRYSOL[48]. The XacGH43_1 crystal structure was fitted into the SAXS molecular envelope using the program SUPCOMB[49].

**Enzyme assays.** XacGH43_1 activity was tested against several natural and 4-nitrophenyl derived substrates (Supplementary Table 1). The 4-paranitrophenyl (pNP) derived substrates were purchased from Sigma-Aldrich (St. Louis, MO) and polymeric substrates were purchased from Megazyme (Wicklow, IE). Reactions with pNP-derived substrates were initiated by the addition of the enzyme and interrupted after specific time intervals by adding a saturated sodium tetraborate solution, followed by heating in 65 °C for 5 min. The hydrolysis of polymeric substrates was evaluated by estimating the reducing sugar released, according to the 3,5-dinitrosalicylic acid method[50]. Initial activity tests were carried out with 2.5 mg mL$^{-1}$ (polysaccharides) or 2.5 mM (synthetic) of substrate, 40 mM mM McIlvaine buffer[51] pH 7.0 and 25 µg of enzyme (6.25 µM). Reactions with pNP-derived substrates were stopped after 1 h and reactions with polymeric substrates after 4 h. Spectrophotometric data were collected in an Infinite® 200 PRO microplate reader (TECAN Group Ltd., Männedorf, Switzerland) using the i-Control software (1.10.4.0) (TECAN).

The optimum pH was evaluated in 40 mM citrate/phosphate/glycine buffer, ranging from 2.5 to 9.5 at 40 °C using 1 mM pNP-β-xylopyranoside (pNP-β-Xyl) and 1 µg (250 nM) of enzyme, with 10 min of reaction. Optimum temperature was evaluated at same conditions above at pH 7.0, from 5 to 60 °C. For the assays with different cations and chelating agents, 5 mM of the respective additive was used in reaction in 40 mM Hepes pH 7.0 at 40 °C with reaction time of 10 min. The same parameters were maintained, with exception of the salt concentration, for testing different CaCl$_2$ concentrations. Kinetics experiments using pNP-β-Xyl as substrate were performed at 40 °C with 0.5 µg (125 nM) of enzyme for 10 min in 40 mM Hepes pH 7.0 with 6 mM CaCl$_2$. For enzyme assays against increasing concentrations of xylan, it was used 25 µg (6.25 µM) of enzyme in 40 mM Hepes pH 7.0 supplemented with 6 mM CaCl$_2$. The temperature was set to 40 °C and a reaction time of 120 min.

To analyze cleavage patterns by CZE, reactions with different XOS from X2 to X6 were carried out at pH 7.0 at 40 °C, using 1 µg (0.32 µM) of enzyme and 5 mM of substrate. Aliquots of each reaction were taken at 0 or 30 min, heated to 95 °C for 15 min and dried. For labeling, samples were incubated with 1 M sodium cyanoborohydride and 6.5 mM APTS for 90 min at 60 °C. Samples were resuspended in running buffer (40 mM potassium phosphate pH 2.5) and analyzed in a P/ACE MDQ instrument (Beckman Coulter, Brea, CA) equipped with a laser-induced fluorescence detection module. A capillary of 50 µm internal diameter and 12 cm effective length was used. Electrophoretic conditions were set to 20 kV mA with reverse polarity at 25 °C. The electrophoretic behavior of labeled products was compared to standards. CZE data were collected using the software 32 Karat 8.0 (Beckman Coulter).

The kinetic experiments of XOS were performed on a Waters Synapt HDMS (Waters Corp., Milford, CT) at V mode and ESI(+) with a spray voltage maintained at 3.0 kV and heated to 130 °C in the source[52]. Data were collected using the software MassLynx (4.1) (Waters). The samples were injected in scan mode ($m/z$ 150−1100) with direct infusion at a flow rate of 20 µL min$^{-1}$ and 1 Hz acquisition. An internal standard with ionization similar to analytes (mannotetraose—M4) was used to increase the accuracy of the method. To ensure that the reaction velocity is equal to the initial velocity, linearity tests were carried with different enzyme concentrations, for X6 and X2 (at 1 mM) in the presence and absence of CaCl$_2$ with different reaction times (0, 2.5, 5, 10, 20, and 30 min). The reaction time and enzyme concentration were selected in a linear region of the hydrolysis reaction. For reactions with X6 in the presence of CaCl$_2$, it was used 2.5 µg mL$^{-1}$ (62.5 nM) XacGH43_1, whereas 5 µg mL$^{-1}$ (125 nM) of the enzyme was employed in reactions without the addition of CaCl$_2$. For reactions with X2, we used 15 µg mL$^{-1}$ (375 nM) in all the reactions. The reactions were set with increasing concentrations of oligosaccharides (X6 and X2) in 30 mM Hepes buffer at pH 7.0, with or without 6 mM CaCl$_2$. After 5 min of reaction, 40 µL methanol were added to stop the enzyme activity. A 15 µL aliquot of the stopped reaction was added with 2 µL mannotetraose (1 mM) and 183 µL water and injected into the mass spectrometer. A calibration curve was obtained to determine the concentrations of the reaction products, plotting the intensity data of the reaction product divided by the internal standard intensity ($I_{X1}/I_{M4}$) versus the xylose concentration (Supplementary Fig. 11). Thus, for enzyme kinetics reactions, the values obtained from intensity $I_{X1}/I_{M4}$ were converted to xylose concentration. The kinetic parameters ($k_{cat}$ and $K_M$) were calculated using OriginPro (OriginLab Corporation, Northampton, MA). All quantitative enzyme assays were expressed as mean ± SD from three independent experiments.

**X-ray crystallography**. XacGH43_1 at 28 mg mL$^{-1}$ (0.7 mM) crystallized by vapor diffusion method in sitting and hanging drops containing 0.2 M ammonium sulfate, 30% (w.v$^{-1}$) polyethylene glycol 8000, 0.1 M sodium cacodylate pH 6.5 and 10% (v.v$^{-1}$) glycerol. To obtain sugar complexes, the crystal was transferred to a solution containing the mother liquor added to 10 mM of sugar solution and incubated at room temperature before being flash-frozen in the nitrogen stream at 100 K and exposed to X-ray radiation.

X-ray diffraction data were collected at the MX2/LNLS beamline (Campinas, São Paulo, Brazil) with a 1.459 Å wavelength, using a PILATUS2M detector (Dectris, Baden-Dattwil, Switzerland) and MXCube software[53]. Diffraction data were scaled and reduced using XDS[54]. The structure of XacGH43_1 was solved by molecular replacement using the program MOLREP[55] and coordinates of the β-xylosidase RS223BX[17] (PDB ID 4MLG) as search model. Two molecules were found in the asymmetric unit and the generated model was refined using phenix. refine[56] and REFMAC5[57], with visual inspections and manual building using COOT[58]. Final model was validated using MolProbity[59] and CheckMyMetal server[60]. Figures containing crystallographic coordinates were generated using Pymol (Schrödinger, LLC, New York). Data processing and refinement statistics are summarized in Supplementary Table 4.

**Molecular docking**. Molecular docking calculations were carried out using Autodock Vina software[61]. To prepare both the protein and the ligand (XacGH43_1 and X6, respectively), it was employed the Autodock Tools graphical interface[62]. The enzyme atomic coordinates were taken from the XacGH43_1 crystal structure, whereas the X6 ligand from PDB entry 4HK8. The surface area evaluated corresponds to the active site pocket and its extended vicinity compatible with the ligand size (a single grid box of $60.0 \times 90.0 \times 60.0$ Å$^3$, centered at $x = 0.5$ Å, $y = 6.5$ Å, and $z = 9.0$ Å). While the enzyme was described as rigid, the ligand was able to rotate each glycosidic bond. Sugar rings were considered as a rigid $^4C_1$ chair conformation, consistent with the X-ray and MD results obtained in this work. The exhaustiveness of search parameter was set at 10, the energy range at 10 kcal mol$^{-1}$, and the desired number of poses up to 20. This calculation was repeated 10 times starting with different random seeds in order to generate a total number of 200 binding poses.

**Classical molecular dynamics simulations**. The initial structure of XacGH43_1 for the MD simulations was taken from the X-ray structure reported in this work (complex XacGH43 + X3), considering the $^2S_O$ conformation of the active site xylosyl unit (nevertheless, during the simulation, the sugar sampled several conformations, as shown in Supplementary Fig. 7b). The missing loops in the crystal structure were constructed with Modeller[63]. All crystallographic water molecules were retained and extra water molecules were added to form a 15 Å water box around the protein surface. For neutralizing the system charge, 13 sodium ions were added to the crystal structure with calcium (15 for the system where the calcium was replaced by a water molecule). The protonation state of all residues was assigned using PROPKA[64], considering pH 7.0. The protonation of His residues was further assessed according their chemical environment. Specifically, His residues 50, 71, 179, 246, 248, 262, 294, and 328 were considered neutral with their proton located at $N_\delta$; His residues 34, 60, 73, 84, 161, 185, 203, 222, 231, 289, 303 and 313 were protonated at $N_\epsilon$ and His25 was double protonated. The general acid (Glu237) was protonated (pKa ≈ 10 according to PROPKA[64], consistent with its role as proton donor during the reaction).

MD simulations were performed using Amber18[65]. The protein was modeled with the FF14SB force-field[66], the X3 with the GLYCAM06 force-field[67], and water molecules were described with the TIP3P force-field[68]. For MD, the systems were initially minimized, keeping both the protein and substrate fixed. Then, the entire systems have been allowed to relax. Afterwards, the system was heated gradually to 100, 200, and 300 K in the NVT ensemble at intervals of 50 ps. Spatial restraints to the protein and ligands were applied during the first heating interval, while all restraints were released after reaching 100 K. Subsequently, the density was converged up to water density at 300 K during 100 ps in the NPT ensemble, followed by a short MD of 100 ps in the NVT ensemble with a time step of 1 fs. The time step was then increased to 2 fs, employing the SHAKE algorithm[69] and the simulation was extended to 100 ns. We launched a total of three independent MD runs distributing random initial velocities during the first heating step. Analyses of the trajectories were carried out using standard tools of Amber and VMD[70]. RMSDs evolutions of the 100 ns production runs are represented in the Supplementary Fig. 12. For analysis of sugar geometrical parameters, we used a documented script[71].

**QM/MM molecular dynamics simulations**. One snapshot from the classical MD-equilibrated systems was taken for subsequent QM/MM calculations at the DFT level of theory. This snapshot had the xylosyl sugar in a $^2S_O/^{2,5}B$ conformation. The QM/MM calculations were performed using CPMD 3.15.1 (IBM Corp. and Max Planck Institute, Stuttgart, Germany) applying the method developed by Laio et al.[72]. that combines Car–Parrinello MD with force-field MD methodology. For the simulation of the conformational FEL, the QM region was composed by the xylosyl at the −1 subsite and half of the sugar at the +1 subsite, whose C2 and C4 were saturated with capping hydrogens. All other atoms of the

systems were included in MM region. Kohn−Sham orbitals were expanded in a planewave basis set with a kinetic energy cutoff of 70 Ry. Norm-conserving Troullier−Martins ab initio pseudopotentials[73] were used for all elements and calculations were performed using the Perdew, Burke, and Ernzerhoff generalized gradient-corrected approximation[74]. This functional form has been proven to give a good performance in the description of hydrogen bonds[75,76] and sugar conformations[77] and was already used with success in previous works on GHs and glycosyltransferases[76,78,79]. A fictitious electron mass of 700 a.u. and a time step of 0.12 fs was used. The structure was optimized using QM/MM MD with annealing of the ionic velocities, until the maximal component of the nuclear gradient was lower than 10$^{-4}$ a.u. Afterwards, the system was re-equilibrated with 6 ps of QM/MM MD at 300 K. A representative snapshot was used as starting points for the corresponding QM/MM metadynamics calculations of the conformational FEL.

**Metadynamics of conformational free energy landscape (FEL)**. Metadynamics[80,81] was used for obtaining the conformational FEL of the −1 sugar in the active site of XacGH43_1. The Cartesian projections of the Cremer–Pople puckering coordinates[33] were used as CVs ($q_x$ and $q_y$)[32,82]. The well-tempered metadynamics approach[83] was used within the metadynamics driver provided by the Plumed2 plugin[84]. The height and the width of the Gaussian terms were set to 1.0 kcal mol$^{-1}$ and 0.1 Å, respectively, for $q_x$ and $q_y$, with a deposition time of 200 MD steps (24 fs) and a bias factor of 10. The simulation was stopped once no quantitative changes in the FEL were observed, specially concerning energy differences among the two main minima. A total number of 2000 Gaussians were deposited. FEL representation was plotted with the program Gnuplot[85].

**Metadynamics simulations of the chemical reaction**. To model the chemical reaction, the QM region was extended to include, beside the sugar atoms, the catalytic residues sidechains and also the catalytic water (Supplementary Fig. 13). Monovalent pseudopotentials, located at the $C_\beta$ atoms of Glu237 and Asp30 and at C3 and C5 of the +1 xylosyl, were used to saturate the QM region[86]. Two metadynamics simulations were performed, starting from each of the two minima of the conformational FEL of the xylosyl unit, corresponding to the $^2S_O/^{2,5}B$ and $^4C_1$ conformations, respectively.

To drive the chemical reaction, we used three CVs that consider all covalent bonds that are to be broken or formed during the reactive process. The first CV (CV$_1$) was taken as the difference between H$_{wat}$–O$_{wat}$ and O$_{Asp237}$–H$_{wat}$ distances (wat = catalytic water). The second CV (CV$_2$) was taken as the difference between C1$_{xylosyl}$–O1$_{xylosyl}$ and O$_{wat}$–C1$_{xylosyl}$ distances, while the third CV (CV$_3$) was taken as the difference between H$_{Glu237}$ –O$_{Glu237}$ and O1$_{xylosyl}$– H$_{Glu237}$. The height and the width of the Gaussian terms were set to 1.0 kcal mol$^{-1}$ and 0.15 Å, respectively, for the three CVs, according to their oscillations observed in a free dynamics. The deposition time was set at 350 MD steps (42 fs). The simulations were stopped after all minima of the FEL were sampled, which involved 1673 and 1563 Gaussians, for reactions starting from the $^2S_O/^{2,5}B$ and from $^4C_1$ substrate conformations, respectively. In terms of metadynamics simulation time, this corresponds to 70 and 66 ps, respectively. Transition states were further refined by isocommittor analyses, according to the following protocol. A region around the point of maximum energy of the reaction pathway[87] in the reaction FEL was defined as a starting point for isocommittor analyses[88,89], which were performed on configurations/structures corresponding to metadynamics frames falling on this region (a total of 23 configurations were initially tested, considering both TS configurations). Twenty independent unbiased dynamical trajectories with random initial velocities were launched for selected configurations, which were stopped once the system reaches either the reactants or the products state. The putative TS for each reaction pathway was taken as the configuration that gives a reactants/products ratio close to 50% (Supplementary Fig. 14).

To further confirm the values of the computed energy barriers, we launched metadynamics simulations with different Gaussian heights (0.50 and 1.00 kcal mol$^{-1}$, for reaction starting from $^4C_1$ and 0.75 and 1.00 kcal mol$^{-1}$, for reaction starting from $^2S_O$) following the protocol proposed in the literature[90], successfully applied to glycosidases[76]. Each simulation was initiated from the point in which ~50–60% of the reactant basin of our previous simulation was filled. The molecular mechanism remained the same independently of the Gaussian height used and the reaction starting from $^4C_1$ consistently showed slightly higher energy barrier values, for all Gaussian heights tested. These simulations resulted in free energy barriers of 12.7 and 15.4 for simulations starting from $^2S_O$ and 18.6 and 16.4 for simulations starting from $^4C_1$.

**Reporting summary**. Further information on research design is available in the Nature Research Reporting Summary linked to this article.

## Data availability

Coordinates and structure factors of XacGH43_1, XacGH43_1 with xylose and XacGH43_1 with xylotriose have been deposited in the Protein Data Bank with accession codes 6XN0, 6XN1 and 6XN2 respectively. Source data are provided with this paper. Additional data supporting the findings of this study are available from the corresponding authors on reasonable request.

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

## Acknowledgements

This research was funded by São Paulo Research Foundation (FAPESP, #2015/26982-0 to M.T.M), Conselho Nacional de Desenvolvimento Científico e Tecnológico (CNPq, #306135/2016-7 to M.T.M.), the Spanish Ministry of Science and Innovation (MICINN, AEI/FEDER, UE) (CTQ2017-85-496 to C.R.), the Spanish Structures of Excellence María de Maeztu (MDM-2017-0767 to C.R.), and the Agency for Management of University and Research Grants (AGAUR) (SGR2017-1189 to C.R.). M.A.B.M. received FAPESP postdoctoral fellowships (#2016/19995-0 and #2018/22138-8) and J.C. received a MICINN predoctoral fellowship FPI-BES-2015-072055. We thank the Brazilian Synchrotron Light Laboratory (LNLS) for the provision of time on the SAXS1 and MX2 beamlines, the Brazilian Biosciences National Laboratory (LNBio) for access to the crystallization (Robolab) facility and the Brazilian Biorenewables National Laboratory (LNBR) for access to the Characterization of Macromolecules (MAC) facility and all their staff for the support. We thank the Red Española de Supercomputación (RES) and the Barcelona Supercomputing Center (BSC) for provision of time in MareNostrum4 and CTE machines (QSB-2020-2-0006), and the staff for technical assistance. We also acknowledge Vanesa P. M. Martins for the help in XacGH43_1 cloning and all members of the groups led by Rovira (Barcelona, Spain) and Murakami (Campinas, Brazil) for the valuable scientific discussions.

## Author contributions

M.A.B.M. performed experiments, solved the structures, and wrote the manuscript. M.A. B.M. and J.C. designed all simulations, analyzed and wrote the results. M.N.D. and J.B.L. C. performed activity assays and analyses. R.A.S.P. and F.C.G. performed mass spectrometry kinetics assays and analyses. C.R.S. and C.C.C.T. performed cloning, expression, purification, and crystallization trials. M.T.M. and C.R. coordinated the project, analyzed results, and wrote the manuscript.

## Competing interests

The authors declare no competing interests.

## Additional information

**Peer review information** *Nature Communications* thanks Mauro Boero, Jan-Hendrik Hehemann, and Alessandro Silva Nascimento for their contributions the peer review of this work. Peer review reports are available.

