## [Peer Review File · Nature Communications]

REVIEWER COMMENTS

Reviewer #1 (Remarks to the Author):

The authors present in this manuscript a thorough and rigorous combined experimental and computational study on one mechanism of action of GH43 xylanolytic enzymes. These are important biotechnological targets, used for hemicellulose degradation in the production of biofuels and extending even beyond this domain, including wastewater treatment (see for instance *Biotechnology for Biofuels* 10, Art. N.: 158 (2017). DOI: 10.1186/s13068-017-0840-y) and agricultural residues disposal and conversion (see *Advances in Life Sciences* 2016, 6(1): 1-6 DOI: 10.5923/j.als.20160601.01). As such, the topics are of worldwide importance and of broad interest in next-generation fuel supply and environment preservation. Moreover, these results are also prone to trigger and inspire engineering applications paving the route for innovative applications of glycoside hydrolases.

To date, the mechanism of most GH43 remains surprisingly unexplored. In this respect, this work is timely and useful to a broad spectrum of scientists and researchers in both academy and industry. A major figure of merit is the accurate atomic-molecular description of the relation between conformation and catalysis in the context of exo-acting GHs. This has so far only been the target of speculation with little or no support from atomistic simulations. It is well known that some sugar conformations promote a catalytic activity and, in fact, authors from this same group have pioneered the field and provided an accurate insight into these conformational changes in the GH families. Here the authors make an important step forward showing that one single enzyme can realize two alternative pathways, accompanied by two different transition states, for a given substrate. This is unprecedented and deserves publication in this Journal in the opinion of this reviewer.

In the use of metadynamics, authors resort to a large number of collective variables for an enzymatic reaction. This large dimensionality, although difficult to explore, is strictly necessary for an accurate sampling of the inverting mechanism, which involves multiple bond cleavage and formation than the more conventional and widely studied retaining mechanism. The manuscript is very well written and logically organized and the extensive set of experimental techniques used for the characterization of the enzyme is a remarkable advantage that, indeed, leads to the discovery of a new activity and provides a new insight into the complex of the wild-type enzyme with an hydrolysable substrate. The general picture provided by this synergy of experiment and simulation is a novel idea on the co-existence of two different catalytic pathways that can be operative for GH43. The crystal structure shows two conformations, supported by the theoretical inspection of the conformational free energy landscape of the -1 xylosyl sugar in the enzyme active site, unambiguously establishing that the enzyme is designed to host only two substrate conformations. Subsequent QM/MM simulations, empowered by free energy sampling techniques (metadynamics), show that each of these conformations can lead to the products of the reaction in two distinct conformational catalytic itineraries. The distinct transition states identified are then further checked via isocommittor analyses, something that is hardly done in enzymatic reactions and makes this work above the average level of standard static QM/MM studies.

For all the reasons summarized in this report, this reviewer reiterates that this manuscript meets all the criteria of quality, importance and scientific level for publication in *Nature Communications*.

Prior to publication, I have to ask authors to consider the following points, which could further improve the manuscript. Specifically:

1. The two GH43 catalytic pathways are close in energy, but the one that initiates from a distorted conformation is favored. Nonetheless, the energy difference turns out to be just 3 kcal/mol. Such a value is not on the verge of the DFT error bar (see the milestone work of Pople and coworkers in *J. Chem. Phys.* 98, 5612 (1993) DOI: 10.1063/1.464906) and, as such, it is reliable. I think the results would be further reinforced if the authors could check how the energy difference depends on the height of the Gaussian biasing functions used in the metadynamics simulations. This does not imply to repeat the full simulations (which would be extremely expensive with 3 CVs), one can start from a

point in which the reactants basin is $\frac{1}{2}$ or $\frac{3}{4}$ filled, in the spirit of the work by the group of D. Marx in J Am Chem Soc. 130, 14148 (2008) DOI: 10.1021/ja802370c.

2. It would be useful to the general readership unfamiliar with metadynamics methods to illustrate the collective variables used in the simulations as an inset in one of the manuscript figures (for instance in Figure 8b). As this figure shows a 2D free energy landscape, it is easy to be confused with a 2D metadynamics (with 2 CVs), whereas in reality 3 CVs have been used.

Reviewer #2 (Remarks to the Author):

The manuscript entitled "Two distinct catalytic pathways for GH43 xylanolytic enzymes unveiled by X-ray and QM/MM simulations" submitted by Morais, Murakami and coworkers uses a very diverse set of experimental and computational tools to uncover (i) the crystal structure of the enzyme XacGH43_1 in a Michaelis complex with xylotriose (X3), (ii) the calcium-dependent activity for this enzyme, with further structural investigations of the role of the calcium ion; (iii) a computational investigation of the reaction mechanism in this GH43 enzyme using QM/MM and Metadynamics; and (iv) the investigation of the most probable reaction pathway from QM/MM calculations.

The manuscript is very well written, and I also acknowledge the amount of experimental work that is shown by the authors in order to strength their hypothesis. It was a real pleasure to read this manuscript.

As the major issue I can point for the manuscript as shown is the lack of novelty. As I will point out below.

1. Matsuzawa and coworkers (J. Biochem. 2017;1-9 doi:10.1093/jb/mvx012) showed back in 2017 that a GH43_1 enzyme from compost metagenome:

* had beta-xylosidase activity (as well as alpha-arabinofuranosidase activity) with a shallow pocket that could accommodate only one xylose in the -1 subsite. They solved a few crystal structures and showed the +1 and +2 subsites with D-Xylp-beta-1,4-D-Xylp-beta, while the subsite -1 showed an alpha-D-Xylp (product). So, my view is that the non-reducing xylose-releasing exo-oligoxyylanase is not a "new activity in these enzymes" as the authors claim (Abstract, lines 35-39).

* had a Ca²⁺ dependent activity, with marked decrease in Km for increasing Ca²⁺ concentrations and consequent increase in catalytic efficiency (kcat/km), very similar to what is shown here for XacGH43_1. The role of the Ca²⁺, its interaction with His288 (XacGH43_1 numbering) are also discussed there. In addition, those authors show in some of their crystal structures (experimental evidence, not a computational model) that the Ca²⁺ ion can be replaced by a Na⁺ ion and a water molecule with the His288 now interacting with the water molecule and not with the Na⁺ ion. So, although the authors extend their analysis to show the effect of the Ca²⁺ ion in melting temperature and in the SAXS data profile, there is some lack of novelty here as well.

2. In the same direction, Barker and coworkers (J. Phys. Chem. B 2010, 114, 15389–15393) showed a very similar investigation of the reaction pathway for the GH43 beta-xylosidase from Geobacillus stearothermophilus using a Car-Parrinello scheme and also Metadynamics. The results obtained are very similar to the results observed in the current manuscript. Of course, there are some methodological improvements when compared to the 10years old paper, but the results remain very similar. So, I also see some lack of novelty here as well.

Something that is intriguing is that Barker and coworkers found the 5S1 conformation as the most stable conformation of the product in their QM/MM investigation. Curiously, the alpha-D-Xylp observed in the -1 subsite in the paper from Matsuzawa (after hydrolysis) is also in the 5S1 conformation. Here,

the authors seem to have find the same conformation, according to what is shown in the figures, but they write 1S5. This should be checked.

A minor issue:

* Page 5, line 123: The authors mention that there is an 8-fold increase in k_{cat}/K_M for X6 hydrolysis as compared to X2. Why is that? The GH43 binding pocket is not deep, but quite shallow. How can the enzyme exhibit a stronger binding for this longer substrate? Any comments on that?

My overall impression is that the manuscript, although very well written and very well supported by experimental and computational data, looks more like a confirmatory result of what is already in the literature and does not bring novelty for the field.

Reviewer #3 (Remarks to the Author):

This is a very nice paper describing the analysis of a glycoside hydrolase of family 43 (GH43) subfamily 1. The paper combines crystallography and modeling as well as experimental kinetics to investigate how this enzyme catalyzes removal of xylose from xylans in an exo-mode of action. Xylan is one of the most abundant polymers in the biosphere, therefore understanding how it is turned over by microbial enzymes is relevant for biotech and ecology. The authors describe the mechanism of hydrolysis and substrate recognition in much detail and discovered multiple interesting and surprising aspects that are to the best of my knowledge new to science. Calcium is a key cofactor in GH42 with 10 fold higher activity due to improved substrate binding. The authors obtained a rare Michaelis complex (transition state mimic) with the native enzyme. This is a rare achievement (normally the substrate is too rapidly broken down for visualization) and in itself very valuable for the glyco field. The authors also found that this enzyme uses two ways to twist the substrate to make it accessible to the nucleophile and the attacking water. This conformational promiscuity is new and therefore important for our basic understanding of these enzymes. If more GH enzymes can twist their substrates in not only one but two ways than this may also influence how one would design inhibitors for GHs involved in disease for example. At least from my perspective these are discoveries of general relevance.

General impression: Very interesting paper, well written, very good experimental data including the Michaelis complex of a native structure, modeling, quantitative data.

Minor points:

The Michealis complex of a native (non-mutated) enzyme is highly significant and should be more prominently featured in the abstract. Maybe like this: ".....the non-reducing xylose-releasing exo-oligoxylanase. We report the first crystal structure of the active GH43 enzyme in a Michaelis-complex with its substrate before hydrolysis."....

The last sentence of the abstract is a bit underwhelming. I think the strength of this study lays in its basic research insight, not in the possible application for finding new xylanases. I would also argue that this conformational flexibility of the sugar ring in the -1 subsite, which was here observed for the first time, might be present in other GH families? If this phenomenon is more common than it is important for how we think about GH substrate or GH inhibitor interaction in general. Therefore, I suggest the authors could broaden their conclusion and highlight its basic research relevance for the CAZyme glyco field.

On a similar note, the significance/relevance of the observation of conformational flexibility in the -1 subsite is not explained well enough. The authors should explain that the current paradigm is that GH stabilizes the conformation that allows engagement with catalytic residues and the water. This stabilization of the transition state is often referred to as induced fit. The observed flexibility of the sugar ring in the -1 subsite is therefore new and could maybe be even called a paradigm shift.

I think the discussion is a bit repetitive with the results section, it reads like a summary about what we already know from reading the results and introduction. Some points that may be interesting to discuss: Why is this not more common? What is the trade-off? Is this conformational flexibility tied to substrate arabinose/xylose flexibility? Are these enzymes less efficient than more specific enzymes?

Line 40: why are the two pathways competitive?

Line 41: I would remove "very", different is different

Line 43: Can such catalytic conformational promicuitivity only extended to other xylanases or also to other enzymes? I think this could be an important discussion point.

Line 78: maybe remove "since long", does not add much.

Line 101: Final sentence of Intro: No need for a reference with this statement. "and adds to the increasing body of mechanistic discoveries in glycosidase catalysis" This is a bit underwhelming and obvious. I also find the fact that this was here described for the first time not very interesting or noteworthy. I think it is more noteworthy that the GH can use two conformations but what does this mean for the CAZyme field, or why is this different to what we knew before? What is the consequence of this discovery?

This section is a bit out of place in the experimental crystallography part.

"Two distinct conformations of the saccharide at the -1 subsite". I would push it down in the manuscript and merge it with the later section: "Two putative catalytic pathways for the same substrate". This way the crystallography and the modeling are better separated. You could either already mention the two crystallographic conformations in the crystallography part or you mention them later in the modeling part for the first time before directly going into the modeling. This would also explain why the modeling was such an important part of this study, to verify the significance of the crystallographic observation.

Line 346: "The fact that a non-preactivated conformation leads to a feasible itinerary could be surprising a priori." Some information is missing here to understand the argument. Please expand on your line of reasoning.

Figure 3: Please clarify if the ligands 2Fo-Fc electron density maps represent maps before or after refining with the ligand. This is important because the shown maps do not reveal the two occupations of the S-1 ring I guess because they are before refinement. It might also be interesting to show the two conformations with the map after refinement.

Material and methods:

Please write not only how much enzyme was used in $\mu\text{g/ml}$ final reaction concentration but also the molarity. This should be done for all enzyme assays.

Please revise grammar and typos in this section. A few are listed but there may be more.

Line 450: ...we used 5 μg ...

Line 451:...we used 15...

Line 452: instead of "that solution" please say what you add to water so that the reader does not have to go to the last sentence.

Line 447-Line448: More detail and explanation is needed: Please provide the substrate concentrations that were used to measure the initial rates. Also please explain how you calibrated the masspec to get from peak integrals to concentrations and show calibration curves in the SI.

Line 456-458: How do you know that quenching was performed in the linear part of the reaction? Please define quenching.

Line 462: polyethylene glycol

Line 464: should read: "added to"

REVIEWER COMMENTS

Reviewer #1 (Remarks to the Author):

The authors present in this manuscript a thorough and rigorous combined experimental and computational study on one mechanism of action of GH43 xylanolytic enzymes. These are important biotechnological targets, used for hemicellulose degradation in the production of biofuels and extending even beyond this domain, including wastewater treatment (see for instance *Biotechnology for Biofuels* 10, Art. N.: 158 (2017). DOI: 10.1186/s13068-017-0840-y) and agricultural residues disposal and conversion (see *Advances in Life Sciences* 2016, 6(1): 1-6 DOI: 10.5923/j.als.20160601.01). As such, the topics of worldwide importance and of broad interest in next-generation fuel supply and environment preservation. Moreover, these results are also prone to trigger and inspire engineering applications paving the route for innovative applications of glycoside hydrolases.

To date, the mechanism of most GH43 remains surprisingly unexplored. In this respect, this work is timely and useful to a broad spectrum of scientists and researchers in both academy and industry. A major figure of merit is the accurate atomic-molecular description of the relation between conformation and catalysis in the context of exo-acting GHs. This has so far only been the target of speculation with little or no support from atomistic simulations. It is well known that some sugar conformations promote a catalytic activity and, in fact, authors from this same group have pioneered the field and provided an accurate insight into these conformational changes in the GH families.

Here the authors make an important step forward showing that one single enzyme can realize two alternative pathways, accompanied by two different transition states, for a given substrate. This is unprecedented and deserves publication in this Journal in the opinion of this reviewer. In the use of metadynamics, authors resort to a large number of collective variables for an enzymatic reaction. This large dimensionality, although difficult to explore, is strictly necessary for an accurate sampling of the inverting mechanism, which involves multiple bond cleavage and formation than the more conventional and widely studied retaining mechanism. The manuscript is very well written and logically organized and the extensive set of experimental techniques used for the characterization of the enzyme is a remarkable advantage that, indeed, leads to the discovery of a new activity and provides a new insight into the complex of the wild-type enzyme with an hydrolysable substrate.

The general picture provided by this synergy of experiment and simulation is a novel idea on the co-existence of two different catalytic pathways that can be operative for GH43.

The crystal structure shows two conformations, supported by the theoretical inspection of the conformational free energy landscape of the -1 xylosyl sugar in the enzyme active site, unambiguously establishing that the enzyme is designed to host only two substrate conformations. Subsequent QM/MM simulations, empowered by free energy sampling techniques (metadynamics), show that each of these conformations can lead to the products of the reaction in two distinct conformational catalytic itineraries. The distinct transition states identified are then further checked via isocommittor analyses, something that is hardly done in enzymatic reactions and makes this work above the average level of standard static QM/MM studies.

For all the reasons summarized in this report, this reviewer reiterates that this manuscript meets all the criteria of quality, importance and scientific level for publication in Nature Communications.

Response: We appreciate the positive evaluation and comments regarding our work. We are pleased that Reviewer#1 has appreciated the computational cutting-edge approach employed in this work and how it is supported and corroborated by high-resolution crystallographic and functional data.

Prior to publication, I have to ask authors to consider the following points, which could further improve the manuscript. Specifically:

1. The two GH43 catalytic pathways are close in energy, but the one that initiates from a distorted conformation is favored. Nonetheless, the energy difference turns out to be just 3 kcal/mol. Such a value is not on the verge of the DFT error bar (see the milestone work of Pople and coworkers in *J. Chem. Phys.* 98, 5612 (1993) DOI: 10.1063/1.464906) and, as such, it is reliable. I think the results would be further reinforced if the authors could check how the energy difference depends on the height of the Gaussian biasing functions used in the metadynamics simulations. This does not imply to repeat the full simulations (which would be extremely expensive with 3 CVs), one can start from a point in which the reactants basin is $\frac{1}{2}$ or $\frac{3}{4}$ filled, in the spirit of the work by the group of D. Marx in *J Am Chem Soc.* 130, 14148 (2008) DOI: 10.1021/ja802370c.

Response: We thank you for suggesting these additional simulations. As recommended, we carried out such simulations using different Gaussian heights for both catalytic pathways. Each simulation was initiated from the point in which approximately 50–60% of the reactant basin of our previous simulation was filled. As expected, the molecular mechanism remained the same, independently of the Gaussian height used, and the reaction starting from 4C_1 always showed

higher barriers values for all tested heights (two for each pathway), supporting our conclusions. The simulations resulted in free energy barriers of 14.1 ± 1.5 kcal.mol⁻¹ (simulations starting from ²S₀, Gaussian heights 0.75 and 1.00 kcal.mol⁻¹) and 17.5 ± 1.1 kcal.mol⁻¹ (simulations starting from ⁴C₁, Gaussian heights 0.50 and 1.00 kcal.mol⁻¹). We included this information in the Methodology section of the manuscript (Methods section).

2. It would be useful to the general readership unfamiliar with metadynamics methods to illustrate the collective variables used in the simulations as an inset in one of the manuscript figures (for instance in Figure 8b). As this figure shows a 2D free energy landscape, it is easy to be confused with a 2D metadynamics (with 2 CVs), whereas in reality 3 CVs have been used.

Response: In the new version of the manuscript we included a representation of CVs (1, 2 and 3) in the Fig. 7b and Fig. 8b schemes, so the reader can easily find them in the main text. Moreover, the CVs schemes and the 3D free energy landscapes are represented in the Supplementary material (Supplementary Figs. 8-10) and referred in the main text and in the legends of Figs. 7 and 8 for further details.

Reviewer #2 (Remarks to the Author):

The manuscript entitled “Two distinct catalytic pathways for GH43 xylanolytic enzymes unveiled by X-ray and QM/MM simulations” submitted by Morais, Murakami and coworkers uses a very diverse set of experimental and computational tools to uncover (i) the crystal structure of the enzyme XacGH43_1 in a Michaelis complex with xylotriose (X3), (ii) the calcium-dependent activity for this enzyme, with further structural investigations of the role of the calcium ion; (iii) a computational investigation of the reaction mechanism in this GH43 enzyme using QM/MM and Metadynamics; and (iv) the investigation of the most probable reaction pathway from QM/MM calculations.

The manuscript is very well written, and I also acknowledge the amount of experimental work that is shown by the authors in order to strength their hypothesis. It was a real pleasure to read this manuscript.

Response: We thank you for the comments about the manuscript.

As the major issue I can point for the manuscript as shown is the lack of novelty. As I will point out below.

1. Matsuzawa and coworkers (J. Biochem. 2017;1-9 doi:10.1093/jb/mvx012) showed back in 2017 that a GH43_1 enzyme from compost metagenome:

*** had beta-xylosidase activity (as well as alpha-arabinofuranosidase activity) with a shallow pocket that could accommodate only one xylose in the -1 subsite. They solved a few crystal structures and showed the +1 and +2 subsites with D-Xylp-beta-1,4-D-Xylp-beta, while the subsite -1 showed an alpha-D-Xylp (product). So, my view is that the non-reducing xylose-releasing exo-oligoxylanase is not a “new activity in these enzymes” as the authors claim (Abstract, lines 35-39).**

Response: We thank the reviewer for this observation, which helps us clarifying an important aspect. We agree with the reviewer that the exo-oligoxylanase activity is probably not a specific feature of the GH43 enzyme that we have investigated (*XacGH43_1*). The exo-oligoxylanase activity might be extended to other members of the same family, which is probably the case of the enzyme from Matsuzawa et al., 2017¹. However, no kinetics data with different xylooligosaccharides were provided in previous studies, therefore one could not evaluate whether the enzyme has preference for longer oligosaccharides (oligoxylanase activity) or for xylobiose (xylosidase activity). Based on the activity naming that was previously adopted², the exo-oligoxylanase activity characterizes higher enzyme efficiency on xylooligosaccharides. Moreover, it is worth to note that, some xylosidases might accept longer xylooligosaccharides ($n > 2$), but they do not have a preference for them, therefore they should not be technically described as exo-oligoxylanases. Thus, kinetics data with different xylooligosaccharides are necessary to claim the exo-oligoxylanase activity. Moreover, our structural X-ray data, along with *in silico* results from molecular docking (see the response to the minor issue below), support that longer xylooligosaccharides can also be accommodated by the enzyme. To avoid misunderstandings, we replaced "we describe a new activity in these enzymes" for "we kinetically and mechanistically describe a new activity in these enzymes". We have also extended the text when citing the previous pioneering results by Matsuzawa and coworkers¹.

*** had a Ca^{2+} dependent activity, with marked decrease in K_m for increasing Ca^{2+} concentrations and consequent increase in catalytic efficiency (k_{cat}/k_m), very similar to what is shown here for *XacGH43_1*. The role of the Ca^{2+} , its interaction with His288 (*XacGH43_1* numbering) are also discussed there. In addition, those authors show in some of their crystal structures (experimental evidence, not a computational model) that the Ca^{2+} ion can be replaced by a Na^+ ion and a water molecule with the His288 now interacting with the water molecule and not with the Na^+ ion. So, although the authors extend their analysis to show the effect of the Ca^{2+} ion in melting temperature and in the SAXS data profile, there is some**

lack of novelty here as well.

Response: We agree with the referee that the role of Ca^{2+} had been discussed in previous studies, which also attempted to address the calcium role in the activity of GH43 enzymes. These include De Sanctis et al. (2010)³, our group (Santos et al., 2014)⁴ and Matsuzawa et al. (2017)¹, all of them cited in our work. Although these studies obtained valuable crystallographic snapshots to identify the coordination sphere of the metal ion, they did not provide compelling data to reveal the mechanism behind calcium activation.

Herein, by means of MD simulations it was possible to get insights into this yet elusive mechanism. In addition, we were fully conscious of the possible occupancy of sodium in the calcium-binding site when it is vacant as first described by our group⁴ and then also by Matsuzawa et al., (2017)¹ (both references were included in the manuscript). However, without the simulations, these previous works did not explain the basis of why the enzyme is more active when calcium is present, while the same does not happen with the monovalent cation or water at the calcium-binding site.

In the proposed model by Matsuzawa et al. (2017)¹ based on crystallographic snapshots, the calcium was proposed to modulate the general protonation of the general acid and to promote a structural change in the catalytic pocket. In the light of our simulations, we observed that calcium is critical to maintain/stabilize the productive active site configuration including the pre-activated conformations of the $-I$ xyloside (please see Supplementary Fig. 7). In our model, supported by crystallography and molecular dynamics simulations, the calcium binding prevents an entropic increase of the histidine side chain, restricting its movement, which would favor the interaction with the general base, leading to the disruption of the productive configuration of the active site, including the critical substrate conformation. It differs from the role suggested by Matsuzawa et al. (2017)¹ since our studies show that calcium was critical to preserve the active form of the general base and not the protonation of the general acid. Moreover, the calcium does not promote structural change, but rather avoids it. Therefore, based on these results we think that our mechanism does not represent an increment in previous proposed models, providing a novel understanding of this activation mechanism.

2. In the same direction, Barker and coworkers (J. Phys. Chem. B 2010, 114, 15389–15393) showed a very similar investigation of the reaction pathway for the GH43 beta-xylosidase from *Geobacillus stearothermophilus* using a Car-Parrinello scheme and also Metadynamics. The results obtained are very similar to the results observed in the current manuscript. Of course, there are some methodological improvements when compared to the 10 years old

paper, but the results remain very similar. So, I also see some lack of novelty here as well.

Response: In the paper of Barker and coworkers⁵, it is not mentioned that more than one itinerary was possible. In fact, only the itinerary beginning from the most frequent *-I* sugar conformation (²S₀) in the structure of the xylosidase mutant (D128G) PDB ID 2EXJ was tested by QM/MM simulations. Differently from the previous work, our crystal structure and simulations revealed two energetically equivalent conformations of the saccharide at the enzyme active site (²S₀ and ⁴C₁) and two distinct catalytic itineraries: ²S₀/^{2,5}B → [^{2,5}B]‡ → ⁵S₁ and ⁴C₁ → [^E₃]‡ → ⁴C₁. Both with similar energy barriers (14.1 and 17.5 kcal.mol⁻¹, respectively), which are close to experimental value (≈ 16 kcal.mol⁻¹), thus indicating that both pathways can be adopted by the enzyme. We believe that the fact of identifying two distinct catalytic itineraries for a GH (for the same substrate) is a disruptive part of our work, along with the novelty that the simulation was informed from a native trapped Michaelis complex (without mutations in the enzyme), a rare complex to be observed. To the best of our knowledge, there is only other case of an active Michaelis complex prior to catalysis for a different family, GH1, which exhibits a distinct fold and mechanism of action⁶. It is only the combination of the two methods (high-resolution crystallography and QM/MM simulations) that has allowed us to detect and describe the two itineraries, something that can likely be extended not only to other GH43 members but possibly also to other exo-acting GH families. In the revised version, we have extended our description of previous work by Barker et al. (2010)⁵ to clarify that only one itinerary was described (pages 10-11).

Something that is intriguing is that Barker and coworkers found the 5S1 conformation as the most stable conformation of the product in their QM/MM investigation. Curiously, the alpha-D-Xylp observed in the -1 subsite in the paper from Matsuzawa (after hydrolysis) is also in the 5S1 conformation. Here, the authors seem to have find the same conformation, according to what is shown in the figures, but they write 1S5. This should be checked.

Response: We thank the Reviewer for bringing this interesting observation. Actually, the ¹S₅ wrote in the first version of the manuscript (in lines 269 and 291) was incorrect (our mistake) and should be replaced by ⁵S₁. This has been corrected in the new version of the manuscript and we have extended the text to mention the agreement with the product conformation observed by Matsuzawa et al (2017)¹

A minor issue:

*** Page 5, line 123: The authors mention that there is an 8-fold increase in kcat/km for X6 hydrolysis as compared to X2. Why is that? The GH43 binding pocket is not deep, but quite shallow. How can the enzyme exhibit a stronger binding for this longer substrate? Any comments on that?**

Response: We acknowledge that this issue could be better supported in the manuscript and we reinforced the structural evidence supporting exo-oligoxyranase activity in this new version. We have also performed molecular docking with X6 (please see Supplementary Fig. 2 and last paragraph of the section "A crystallographic snapshot of a native Michaelis substrate complex") to explore the binding of this long substrate. Structural analysis showed that the *XacGH43_1* active site does not impose steric impediments to longer xylooligosaccharides and, interestingly, molecular docking performed with X6 reveal that the N-terminal helix and very C-terminus might serve as a platform to interact with the sugar moieties +3, +4 and +5, which can make both hydrophobic contacts and hydrogen bonds with the enzyme (Supplementary Fig. 2). It is also possible that missing residues at the N-terminus (4 aa) could yet become structured upon the binding of longer xylooligosaccharides, providing further stabilizing interactions, which could not be assessed with current available structures due to the lack of electron density. In addition, the docking results nicely agree with the xylotriose Michaelis complex obtained by X-ray crystallography (subsites -1, +1 and +2) supporting our interpretation. These new results are now discussed in page 6.

My overall impression is that the manuscript, although very well written and very well supported by experimental and computational data, looks more like a confirmatory result of what is already in the literature and does not bring novelty for the field.

Response: We thank again the reviewer for the raised points, and we expect to have addressed all issues, demonstrating the novelty brought by our work, which were also noted and highlighted by the other reviewers. The novel findings in our work are in our opinion the following:

- (1) The first crystal structure of a native inverting GH enzyme in complex with its substrate.
- (2) Demonstration of the reducing-end xylose-releasing exo-oligoxyranase activity.
- (3) Discovery of more than one catalytic itinerary in a GH enzyme, breaking the paradigm that a GH family would have only one energetically favorable catalytic route for a given substrate.
- (4) An atomistic explanation for the essential structural role of the metal ion.

Reviewer #3 (Remarks to the Author):

This is a very nice paper describing the analysis of a glycoside hydrolase of family 43 (GH43) subfamily 1. The paper combines crystallography and modeling as well as experimental kinetics to investigate how this enzyme catalyzes removal of xylose from xylans in an exo-mode of action. Xylan is one of the most abundant polymers in the biosphere, therefore understanding how it is turned over by microbial enzymes is relevant for biotech and ecology.

The authors describe the mechanism of hydrolysis and substrate recognition in much detail and discovered multiple interesting and surprising aspects that are to the best of my knowledge new to science. Calcium is a key cofactor in GH43 with 10 fold higher activity due to improved substrate binding. The authors obtained a rare Michaelis complex (transition state mimic) with the native enzyme. This is a rare achievement (normally the substrate is too rapidly broken down for visualization) and in itself very valuable for the glyco field. The authors also found that this enzyme uses two ways to twist the substrate to make it accessible to the nucleophile and the attacking water. This conformational promiscuity is new and therefore important for our basic understanding of these enzymes. If more GH enzymes can twist their substrates in not only one but two ways than this may also influence how one would design inhibitors for GHs involved in disease for example. At least from my perspective these are discoveries of general relevance.

General impression: Very interesting paper, well written, very good experimental data including the Michaelis complex of a native structure, modeling, quantitative data.

Response: We thank the Reviewer #3 for the comments and the careful review of our work. The comments (below) were very valuable to better present our findings and their impact in the field.

Minor points:

The Michealis complex of a native (non-mutated) enzyme is highly significant and should be more prominently featured in the abstract. Maybe like this: “.....the non-reducing xylose-releasing exo-oligoxyranase. We report the first crystal structure of the active GH43 enzyme in a Michaelis-complex with its substrate before hydrolysis.”....

Response: We thank the Reviewer for this comment. We modified the abstract accordingly.

The last sentence of the abstract is a bit underwhelming. I think the strength of this study lays in its basic research insight, not in the possible application for finding new xylanases. I would also argue that this conformational flexibility of the sugar ring in the -1

subsite, which was here observed for the first time, might be present in other GH families? If this phenomenon is more common than it is important for how we think about GH substrate or GH inhibitor interaction in general. Therefore, I suggest the authors could broaden their conclusion and highlight its basic research relevance for the CAZyme glyco field.

On a similar note, the significance/relevance of the observation of conformational flexibility in the -1 subsite is not explained well enough. The authors should explain that the current paradigm is that GH stabilizes the conformation that allows engagement with catalytic residues and the water. This stabilization of the transition state is often referred to as induced fit. The observed flexibility of the sugar ring in the -1 subsite is therefore new and could maybe be even called a paradigm shift.

Response: Thank you for pointing the high relevance of our findings to basic research in the glycobiology field. Based on these comments, we modified the concluding paragraph of the abstract and also strengthened the discussion section with these points. A new study (just published while this manuscript was under review), from a retaining beta-galactocerebrosidase⁷, a disease-related GH, contributes to this paradigm shift that a glycosidase can operate by two catalytic itineraries. This beta-galactocerebrosidase is also an exo-enzyme, but operates by a retaining mechanism, while the GH43 enzyme operates via an inverting mechanism. Therefore, based on these observations, it is likely that such catalytic conformational promiscuity, coined in this study, might be present in other GH families.

I think the discussion is a bit repetitive with the results section, it reads like a summary about what we already know from reading the results and introduction. Some points that may be interesting to discuss: Why is this not more common? What is the trade-off? Is this conformational flexibility tied to substrate arabinose/xylose flexibility? Are these enzymes less efficient than more specific enzymes?

Response: From our observations (and also better discussed in this new version of the manuscript) the catalytic conformational promiscuity might be present in several exo-enzymes, since this mode of action might be more prone or can easier accept non-distorted sugar conformations. In the case of GH43 and other bi-functional enzymes the open active-site nature further contributes to the accommodation of different substrate conformations, while the orientation of the hydroxyl groups does not vary between the two Michaelis complex with xylooligosaccharides. These observations suggest that some exo-enzymes have evolved to stabilize different pathways, to expand their catalytic possibilities.

Moreover, this concept of catalytic conformational promiscuity does not seem to be strictly linked to the xylose/arabinose, since as discussed above, the retaining exo-galactocerebrosidase can also accept two alternative conformations of the leaving group. It is likely that the mechanistic characterization of other GH families, over more than 170 so far described, will lead to the expansion of this concept. By comparing GH43 kinetics data with other beta-xylosidases/arabinofuranosidases present in the literature, there is no evidence that could point that these enzymes are less efficient. So apparently the catalytically viable non-distorted sugar conformation did not raise at a cost of decreasing enzyme performance. We have expanded the discussion to reflect these observations.

Line 40: why are the two pathways competitive?

Response: It was inferred based on the stability of both conformations of the Michaelis complex, which are thermodynamically possible and exhibit similar energy barriers of the transition states, showing that both itineraries are kinetically accessible, even though one itinerary would be favored. In view of the reviewer comment, we realized this can be confusing and have changed “competitive” by “possible”.

Line 41: I would remove “very”, different is different.

Response: We agree with the Reviewer and it was removed.

Line 43: Can such catalytic conformational promiscuity only extended to other xylanases or also to other enzymes? I think this could be an important discussion point.

Response: Yes, it is reasonable to expect that other GH families beyond those comprising xylanolytic enzymes can exhibit more than one energetically possible catalytic route as timely corroborated by a just published work with the GH59 family (please see the discussion above). The abstract and discussion were modified to highlight this aspect. Other GH enzymes (from other families) are being investigated in our laboratories and we can tentatively predict from our preliminary data that this is a common trend in exo-GHs.

Line 78: maybe remove “since long”, does not add much.

Response: We removed it in this new version.

Line 101: Final sentence of Intro: No need for a reference with this statement. “and adds to the increasing body of mechanistic discoveries in glycosidase catalysis” This is a bit underwhelming and obvious. I also find the fact that this was here described for the first time not very interesting or noteworthy. I think it is more noteworthy that the GH can use two conformations but what does this mean for the CAZyme field, or why is this different to what we knew before? What is the consequence of this discovery?

Response: Glycoside hydrolases comprise a large superfamily with over 170 families and, in a single family, dozens of activities on distinct substrates can be found. On the other hand, the knowledge of the catalytic mechanism of these enzymes is yet very limited compared to the sheer complexity of CAZymes, indicating that this research area is not mature yet. This finding of catalytic conformational promiscuity is new in these enzymes and it is possible that other families would exhibit such catalytic behavior. Prior to this study we were expecting to always find a single or by far most favorable catalytic itinerary starting from a single pre-activated conformation. Here, we showed that catalytic reaction can be initiated by two distinct conformations and occur through two distinct transition states. This finding, according it will be expanded to other families as just done to GH59 as well, will redact how we should look to substrate and inhibitor interaction. Some elements of this discussion were included in the last paragraph of the introduction and throughout the discussion section.

This section is a bit out of place in the experimental crystallography part. “Two distinct conformations of the saccharide at the -1 subsite”. I would push it down in the manuscript and merge it with the later section: “Two putative catalytic pathways for the same substrate”. This way the crystallography and the modeling are better separated. You could either already mention the two crystallographic conformations in the crystallography part or you mention them later in the modeling part for the first time before directly going into the modeling. This would also explain why the modeling was such an important part of this study, to verify the significance of the crystallographic observation.

Response: We thank the Reviewer for the nice suggestion. We re-ordered and merged the sections mentioned and we believe that indeed it reads better in this new version of the manuscript.

Line 346: “The fact that a non-preactivated conformation leads to a feasible itinerary could be surprising a priori.” Some information is missing here to understand the argument. Please expand on your line of reasoning.

Response: Thank you for pointing this lack of completeness in the sentence. It rephrased accordingly and now it reads: "*The fact that a non-preactivated conformation leads to a feasible itinerary could be surprising a priori, since it has been shown that all β -GHs operate via a catalytic itinerary that begins from a specific distorted conformation of the sugar*".

Figure 3: Please clarify if the ligands 2Fo-Fc electron density maps represent maps before or after refining with the ligand. This is important because the shown maps do not reveal the two occupations of the S-1 ring I guess because they are before refinement. It might also be interesting to show the two conformations with the map after refinement.

Response: The maps shown in the manuscript are after refinement and we added a sentence on the legends of Figs 3 and 6 with this information in the new version of the manuscript. We acknowledge that could be casuistical to differentiate these two conformations solely based on electron density maps at 1.65 Å resolution as the difference is subtle and the position of hydroxyl groups at 2S_0 or 4C_1 conformations are maintained (in *XacGH43_1* + X3 complex). In this case, the decision of modeling both conformations was also based on a cross-validation process between QM/MM simulations that showed both conformations were almost equally stable and discrete improvement in the F_c - F_o map (green in Figure R1) when the two conformations were modeled with 50% of occupancy each. Note in the lateral views of figure R1, the improvement in the F_c - F_o maps with both conformations modeled.

Figure R1. Conformations of the -1 xylopyranosyl of *XacGH43_1* + X3 complex, after crystallographic refinement. $2F_o$ - F_c electron density maps are contoured at 1.8 σ and shown in grey. F_o - F_c electron density maps are contoured at 2.5 σ and shown in green. 2S_0 and 4C_1 sugars (two first panels, on left) were modeled with 100% of occupancy. ${}^2S_0 / {}^4C_1$ double conformation (last panel, on right) was modeled with 50% of occupancy for each conformer.

Material and methods:

Please write not only how much enzyme was used in $\mu\text{g/ml}$ final reaction concentration but also the molarity. This should be done for all enzyme assays.

Please revise grammar and typos in this section. A few are listed but there may be more.

Line 450: ...we used 5 μg ...

Line 451: ...we used 15...

Response: For lines 450, 451 and others (where applicable) we replaced the volume and concentration by the total mass, as suggested by the Reviewer. Grammar and typos were double-checked and fixed.

Line 452: instead of “that solution” please say what you add to water so that the reader does not have to go to the last sentence.

Response: We rewrote this entire section in the new version of the manuscript.

Line 447-Line448: More detail and explanation is needed: Please provide the substrate concentrations that were used to measure the initial rates. Also please explain how you calibrated the masspec to get from peak integrals to concentrations and show calibration curves in the SI.

Response: We included more info in the Methods section. We also added the calibration curve in the Supporting Information (please, see Supplementary Figure 11). The substrates concentrations for measuring initial rates were 1 mmol.L^{-1} (for X2 and X6). The calibration curve was built by plotting the intensity data of the reaction product divided by the internal standard intensity (IX1 / IM4) versus the xylose (X1) concentration.

Line 456-458: How do you know that quenching was performed in the linear part of the reaction? Please define quenching.

Response: We used "quenching" to refer to "stopping of enzyme reaction", in our case by denaturing the protein with addition of an organic solvent (methanol). For better clarity, we have rewritten this

part of the methodology and the term "quenching" has been removed. To ensure that the reaction velocity is equal to the initial velocity, linearity tests were carried with different enzyme concentrations, for X6 and X2 (at 1 mM) in the presence and absence of CaCl₂ with different reaction times. The reaction time and enzyme concentration were selected in a linear region of the hydrolysis reaction. For reactions containing X6 as substrate, we used 5 µg.mL⁻¹ (125 nM) of *XacGH43_1* and 2.5 µg.mL⁻¹ (62.5 nM) in presence of CaCl₂ and for reactions containing X2, we used 15 µg.mL⁻¹ (375 nM) in all the reactions. As it can be seen in the Figure R2, the selected time of 5 min is in the linear region, for all reactions.

Figure R2. Linearity tests, showing that in the selected conditions used for enzyme kinetics, the reaction velocity is equal to the initial velocity.

Line 462: polyethylene glycol

Response: We corrected the typo error.

Line 464: should read: "added to"

Response: We corrected the typo error.

REFERENCES

1. Matsuzawa, T., Kaneko, S., Kishine, N., Fujimoto, Z. & Yaoi, K. Crystal structure of metagenomic β -xylosidase/ α -L-arabinofuranosidase activated by calcium. *J. Biochem.* **162**, 173–181 (2017).
2. Honda, Y. & Kitaoka, M. A family 8 glycoside hydrolase from *Bacillus halodurans* C-125 (BH2105) is a reducing end xylose-releasing exo-oligoxylanase. *J. Biol. Chem.* **279**, 55097–55103 (2004).
3. De Sanctis, D. *et al.* *Cellvibrio japonicus* α -l-arabinanase 43a has a novel five-blade β -propeller fold. *J. Biol. Inorg. Chem.* **19**, 505–513 (2010).
4. Santos, C. R. *et al.* Mechanistic strategies for catalysis adopted by evolutionary distinct family 43 arabinanases. *J. Biol. Chem.* **289**, 7362–7373 (2014).
5. Barker, I. J., Petersen, L. & Reilly, P. J. Mechanism of xylobiose hydrolysis by GH43 β -xylosidase. *J. Phys. Chem. B* **114**, 15389–15393 (2010).
6. Isorna, P. *et al.* Crystal Structures of *Paenibacillus polymyxa* β -Glucosidase B Complexes Reveal the Molecular Basis of Substrate Specificity and Give New Insights into the Catalytic Machinery of Family I Glycosidases. *J. Mol. Biol.* **371**, 1204–1218 (2007).
7. Nin-hill, A. & Rovira, C. The Catalytic Reaction Mechanism of β -galactocerebrosidase, the Enzyme Deficient in Krabbe Disease The Catalytic Reaction Mechanism of β -Galactocerebrosidase, the Enzyme Deficient in Krabbe Disease. (2020) doi:10.1021/acscatal.0c02609.

REVIEWERS' COMMENTS

Reviewer #1 (Remarks to the Author):

Remarks on the technical assessment:

The present revised version of the original manuscript addresses all the technical concerns that this and the other Reviewers have brought to the attention of the authors. Specifically, the issues of the dimensionality in terms of collective variables and the error bar affecting any DFT-based estimation of the energetics have found a consistent answer.

Overall Remarks on the manuscript:

After reading the revised manuscript, the supporting information and the specific answers to all the questions and issues raised by all the referees, this reviewer feels that the authors have done a remarkable effort to remove any source of concern. As stated in my original report, the topics presented, supported by both experiments and forefront molecular modeling, of worldwide importance are prone to be of general interest to the broad community of researchers working in next-generation fuels and related environmental impact.

There are no other major concerns from this reviewer.

The manuscript can be deemed as publishable in its present amended form.

Reviewer #2 (Remarks to the Author):

The revised manuscript by Morais and coworkers addressed all the issues I mentioned in my revision of the first version submitted. I thank the authors for their comments. My major concern about the lack of novelty was better explained in their rebuttal letter and was also addressed in the revised manuscript, making it clear the novelty their work brings and their applications for other CAZymes. So, I think all the points raised have been properly addressed.

Alessandro S. Nascimento

Reviewer #3 (Remarks to the Author):

Dear authors,

Thank you for rigorously addressing all of my comments. This is a very fine paper that clearly expands our understanding of the mechanism of CAZymes. The amount of work and the resulting new, exciting discoveries bring the field substantially forward and justify publication in Nature Communications.

Minor points:

These points are just suggestions.

There is a claim of a first GH43 Michaelis complex in the abstract, not sure this being first claim is necessary.

In the abstract it is written that xylan is the second most abundant polysaccharide in nature after cellulose I guess. I think this is an assumption based on tree (plant) counting. However, to the best of my knowledge nobody directly measured the amount of types of polysaccharides in nature on a global

scale. There could be other important polysaccharides that are more abundant. I would qualify this statement to be on the safe side.

Last sentence of abstract is much better than before but I would rather write: "These findings expand the current general model..." Change is too strong for my taste.

Line 598: To further nail down?

Last sentence of the paper seems a bit too strong and could easily be argued with. This may dampen interest in this paper. I would soften the sentence a bit, here is just a suggestion:

The here described conformational promiscuity exemplifies that some glycosidases are more flexible than others in bending their substrate for catalysis. Conformational promiscuity may be relevant for the design of transition state-like inhibitors and activity-based probes for glycoside hydrolases opening new venues in glycobiology.

REVIEWERS' COMMENTS

Reviewer #1 (Remarks to the Author):

Remarks on the technical assessment:

The present revised version of the original manuscript addresses all the technical concerns that this and the other Reviewers have brought to the attention of the authors. Specifically, the issues of the dimensionality in terms of collective variables and the error bar affecting any DFT-based estimation of the energetics have found a consistent answer.

Overall Remarks on the manuscript:

After reading the revised manuscript, the supporting information and the specific answers to all the questions and issues raised by all the referees, this reviewers feels that the authors have done a remarkable effort to remove any source of concern. As stated in my original report, the topics presented, supported by both experiments and forefront molecular modeling, of worldwide importance are prone to be of general interest to the broad community of researchers working in next-generation fuels and related environmental impact.

There are no other major concerns from this reviewer.

The manuscript can be deemed as publishable in its present amended form.

Response: We thank the Reviewer #1 for the careful revision, comments and suggestions throughout the reviewing process.

Reviewer #2 (Remarks to the Author):

The revised manuscript by Morais and coworkers addressed all the issues I mentioned in my revision of the first version submitted. I thank the authors for their comments. My major concern about the lack of novelty was better explained in their rebuttal letter and was also addressed in the revised manuscript, making it clear the novelty their work brings and their applications for other CAZymes. So, I think all the points raised have been properly addressed.

Alessandro S. Nascimento

Response: We thank the Reviewer #2 for the comments and suggestions, which contributed to make clearer the relevance of our work.

Reviewer #3 (Remarks to the Author):

Dear authors,

Thank you for rigorously addressing all of my comments. This is a very fine paper that clearly expands our understanding of the mechanism of CAZymes. The amount of work and the resulting new, exciting discoveries bring the field substantially forward and justify publication in Nature Communications.

Response: We appreciate the comments from Reviewer #3 and the several valuable suggestions to improve our manuscript.

Minor points:

These points are just suggestions.

There is a claim of a first GH43 Michaelis complex in the abstract, not sure this being first claim is necessary.

Response: We followed the Reviewer suggestion. Although, to the best of our knowledge there is no other active GH43 Michaelis complex reported, we agree that the claim of “first Michaelis complex” is not necessary.

In the abstract it is written that xylan is the second most abundant polysaccharide in nature after cellulose I guess. I think this is an assumption based on tree (plant) counting. However, to the best of my knowledge nobody directly measured the amount of types of polysaccharides in nature on a global scale. There could be other important polysaccharides that are more abundant. I would qualify this statement to be on the safe side.

Response: We thank the Reviewer #3 for bringing it up. We modified the referred sentence accordingly.

Last sentence of abstract is much better than before but I would rather write:” These findings expand the current general model...” Change is too strong for my taste.

Response: We agree with Reviewer #3 and we replaced “change” by “expand” in the abstract last sentence.

Line 598: To further nail down?

Response: We replaced “nail down” by “confirm”.

Last sentence of the paper seems a bit too strong and could easily be argued with. This may dampen interest in this paper. I would soften the sentence a bit, here is just a suggestion:

The here described conformational promiscuity exemplifies that some glycosidases are more flexible than others in bending their substrate for catalysis. Conformational promiscuity may be relevant for the design of transition state-like inhibitors and activity-based probes for glycoside hydrolases opening new venues in glycobiology.

Response: We agree with the Reviewer #3 and modified the last sentence taking the suggestion into account.